# Valine aminoacyl-tRNA synthetase promotes therapy resistance in melanoma

Najla El-Hachem [1], Marine Leclercq [1,15], Miguel Susaeta Ruiz[1,15], Raphael Vanleyssem [1], Kateryna Shostak [2], Pierre-René Körner [3], Coralie Capron [4], Lorena Martin-Morales [4], Patrick Roncarati[5], Arnaud Lavergne [6], Arnaud Blomme [1], Silvia Turchetto[7], Eric Goffin [8], Palaniraja Thandapani[9], Ivan Tarassov [10], Laurent Nguyen [7,11], Bernard Pirotte[8], Alain Chariot[2,11], Jean-Christophe Marine[12,13], Michael Herfs [5], Francesca Rapino [4,11], Reuven Agami [3,14] & Pierre Close [1,11] ✉

Transfer RNA dynamics contribute to cancer development through regulation of codon-specific messenger RNA translation. Specific aminoacyl-tRNA synthetases can either promote or suppress tumourigenesis. Here we show that valine aminoacyl-tRNA synthetase (VARS) is a key player in the codon-biased translation reprogramming induced by resistance to targeted (MAPK) therapy in melanoma. The proteome rewiring in patient-derived MAPK therapy-resistant melanoma is biased towards the usage of valine and coincides with the upregulation of valine cognate tRNAs and of VARS expression and activity. Strikingly, VARS knockdown re-sensitizes MAPK-therapy-resistant patient-derived melanoma in vitro and in vivo. Mechanistically, VARS regulates the messenger RNA translation of valine-enriched transcripts, among which hydroxyacyl-CoA dehydrogenase mRNA encodes for a key enzyme in fatty acid oxidation. Resistant melanoma cultures rely on fatty acid oxidation and hydroxyacyl-CoA dehydrogenase for their survival upon MAPK treatment. Together, our data demonstrate that VARS may represent an attractive therapeutic target for the treatment of therapy-resistant melanoma.

Patients with $BRAF^{V600E}$ melanoma represent about 50% of human melanoma[1,2], harbour constitutive MAPK signalling activation and are eligible for MAPK-targeted therapy[3]. While therapeutic response is high (up to 70%), patients experience disease evolution after months of treatment because of acquired drug resistance[4]. In addition to genetic mutations, non-genetic adaptation mechanisms also account for treatment resistance[5,6]. Interestingly, RNA modifications (occurring at messenger RNAs, ribosomal RNAs (rRNA) or transfer RNAs) sustain cancer cells adaptation to stress. For instance, $N^6$-methyladenosine (m6A) modification of specific mRNAs supports melanoma resistance to MAPK therapy, through eIF-4F activity[7,8]. Similarly, the wobble uridine tRNA modification pathway promotes $BRAF^{V600E}$ melanoma resistance through HIF1A mRNA translation in a codon-specific manner[9].

tRNA molecules and aminoacyl-tRNA synthetases (aaRSs) are important in diseases, cancer progression and metastatic formation, presumably to accommodate changes in protein synthesis[10–13]. tRNA aminoacyl transferases are highly conserved enzymes that load amino acids to their cognate tRNA and ensure protein synthesis and translation accuracy. Non-canonical roles of aaRSs vary from transcription regulation, cell migration, angiogenesis and other functions[14]. aaRS loss of function is often associated with neurodegenerative diseases suggesting that the nervous system is particularly sensitive to aaRS and tRNA misregulation[15]. Biallelic mutations of cytoplasmic valine aminoacyl-tRNA synthetase (VARS) are associated with neurodevelopmental epileptic encephalopathy. This correlates with reduced VARS aminoacylation enzymatic activity[16].

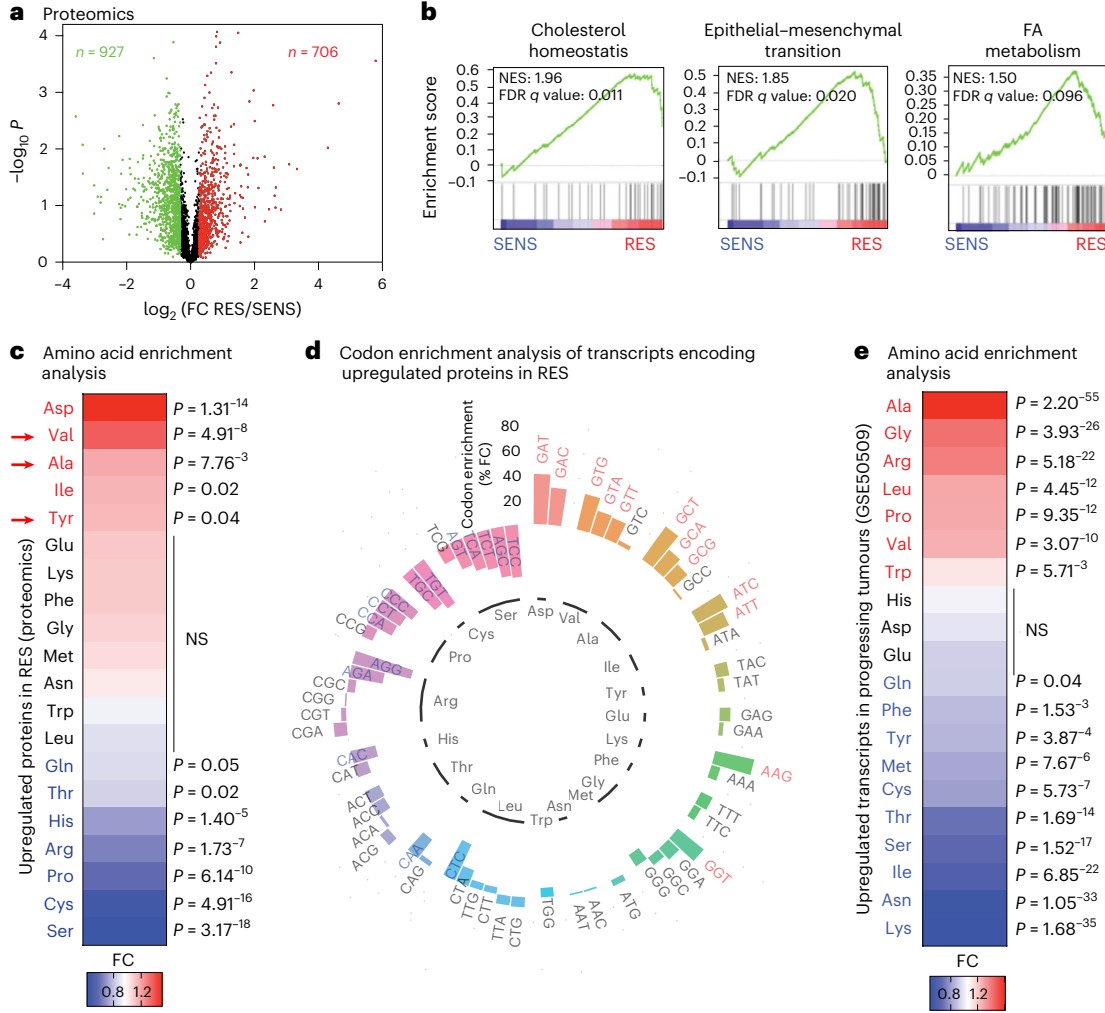

**Fig. 1 | Valine is enriched in proteins upregulated during resistance to MAPK therapy. a**, Volcano plots showing *P* values (−log$_{10}$) of two-tailed unpaired *t*-tests versus the log$_2$ fold change (FC) of the proteomics analysis of melanoma M395 SENS or RES to vemurafenib (*n* = 3); red indicates upregulated and green indicates downregulated proteins (*n* = 3). **b**, GSEA analysis using the Hallmark database. Normalized enrichment score (NES) and false discovery rate (FDR) *q* values are added for each dataset. **c**, A heat map representing amino acids enrichment analysis. **d**, A circular plot representing codon content analysis in transcripts encoding proteins upregulated in M395 RES compared with the genome. **e**, A heat map representing amino acids analysis of mRNA from progressing tumours after treatment (GSE50509). The significance of amino acid and codon enrichment was calculated using a chi-squared test (**c**–**e**). The significant differences are represented; the red indicates an enrichment and blue an impoverishment. NS, not significant.

Recent studies also revealed the implication of aaRS in cancer as potential therapeutic targets. The knockdown of threonine aaRS in pancreatic cancer suppresses mucin-1 levels, an oncoprotein enriched in threonine that favours cancer cell migration[17]. Also, inhibitors of proline tRNA aminoacyl-transferase prevent fibrosis by blocking the biosynthesis of collagen, a proline-rich protein[18].

In this Article, we characterize the translation reprogramming occurring in *BRAF*^*V600E* melanoma upon resistance to MAPK therapy to uncover key mechanisms driving drug resistance. By combining tRNA sequencing, polysome, ribosome profiling and quantitative proteomics, we demonstrate that VARS is a key driver of therapy resistance through the translation of specific mRNA transcripts.

## Results

### Amino acid usage in the proteome of resistant melanoma

Specific reprograming of mRNA translation modulates phenotype switching and therapy resistance in melanoma with no obvious changes in global mRNA translation[8,9,19,20]. Accordingly, we did not observe a significant change in global mRNA translation

in naive (SENS) versus MAPK-therapy-resistant (RES) patient melanoma cultures, as assessed by polysome profiling and in O-propargyl-puromycin (OP-puro) incorporation (Extended Data Fig. 1a,b). Therefore, we hypothesized that mRNA translation of specific mRNAs is dynamically modulated upon resistance to MAPK therapy. We assessed the proteome rewiring occurring in MAPK therapy resistance by quantitative proteomic analyses (Fig. 1a). The top ten gene set enrichment analysis (GSEA) confirmed that cholesterol hoemeostasis, epithelial-to-mesenchymal transition and fatty acid (FA) metabolism pathways were increased in RES cultures, consistent with previous studies[21–23] (Fig. 1b and Extended Data Fig. 1c). First, we performed amino acid enrichment analyses with proteins differentially expressed in RES cultures. Interestingly, aspartate, valine, alanine, isoleucine and tyrosine were statistically enriched in proteins upregulated in RES cultures (Fig. 1c). Only valine, alanine and tyrosine were specifically enriched in proteins upregulated but not in downregulated ones in RES cultures (Extended Data Fig. 1d and Supplementary Table 1). Hence, in-depth analysis of the codon composition of mRNAs encoding upregulated proteins in RES cultures

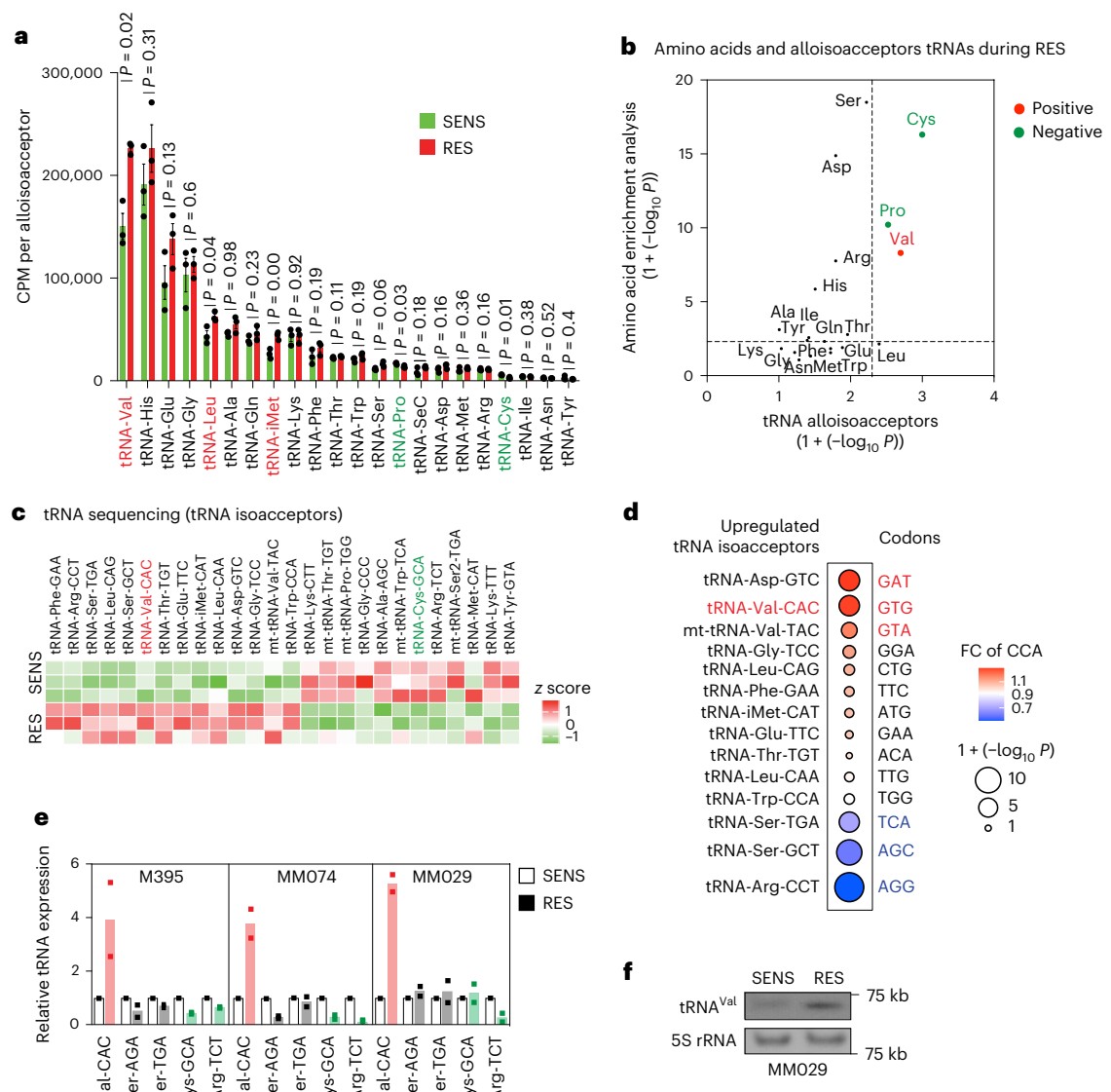

**Fig. 2 | Valine tRNAs are upregulated in resistant melanoma. a**, A histogram showing all tRNA alloisoacceptors in M395 SENS and RES cultures (sum ± s.e.m. of $n = 3$ independent replicates, and a two-tailed unpaired $t$-test was performed); red represents the significantly upregulated tRNAs in RES and green represents the significantly downregulated tRNAs in RES. **b**, Correlation between significant changes in amino acids from the amino acid content analysis (chi-squared test) and the tRNA alloisoacceptors (two-tailed unpaired $t$-test; no adjustment was used for these comparisons). The red represents the enriched amino acids and increase in the corresponding tRNAs, and the green represents impoverished amino acids and corresponding tRNAs. **c**, A heat map representing the $z$ scores of significantly modified tRNA isoacceptors (by tRNA sequencing) in M395 SENS and RES cultures ($n = 3$). The red represents tRNA-Val-CAC as one of the

upregulated tRNAs isoacceptors in RES and the green represents tRNA-Cys-GCA as one of the downregulated tRNAs isoacceptors in RES. **d**, Correlation between significant changes in codons from the CCA (chi-squared test) and the upregulated tRNA isoacceptors (two-tailed unpaired $t$-test; no adjustment was used for these comparisons). The red represents enriched codons and increase in the corresponding isoacceptors tRNAs and the blue represents the impoverished codons and corresponding isoacceptors tRNAs. **e**, Relative expression (FC) of the indicated tRNAs assessed by RT–qPCR analysis (after tRNA demodification) in various SENS and RES cell cultures ($n = 2$). Red, increased expression; green, decreased expression. **f**, Northern blot representing tRNA-Val-CAC in SENS and RES MM029 melanoma cultures. A 5S rRNA is shown for normalization purpose.

($n = 706$; Fig. 1a; codon content analysis (CCA) done as in ref. 24) confirmed that aspartate codons (GAT, $P = 9.00^{-10}$; GAC, $P = 6.17^{-6}$), three of the four valine codons (GTG, $P = 1.06^{-5}$; GTA, $P = 4.56^{-3}$; GTT, $P = 7.76^{-3}$), three of the four alanine codons (GCT, $P = 8.23^{-7}$; GCA, $P = 3.52^{-3}$; GCG, $P = 0.02$) and two isoleucine codons (ATC, $P = 3.50^{-5}$; ATT, $P = 0.001$) were significantly enriched (Fig. 1d and Supplementary Table 1). Finally, similar analyses were performed using a dataset of patients with progressing *BRAF*$^{V600E}$ metastatic melanoma (GSE50509). Among others, valine and alanine were significantly over-represented in genes upregulated in progressing tumours but not in downregulated genes (Fig. 1e and Supplementary Table 2).

**Valine tRNAs are upregulated in therapy-resistant melanoma**

The changes in tRNA expression are presumed to accommodate proteome requirements and support tumour progression and metastasis[10,11,25]. We performed tRNA sequencing in SENS and RES melanoma cultures. Significant differences in the expression levels of tRNAs were observed; among them, the alloisoacceptor valine tRNA was significantly upregulated (Fig. 2a and Extended Data Fig. 2a) and correlated with the enrichment of valine amino acid reported in RES cultures (Fig. 2b). Particularly, the tRNA-Val-CAC isoacceptor was significantly upregulated (Fig. 2c) and correlated with the specific enrichment of the cognate GTG codon in transcripts encoding upregulated proteins

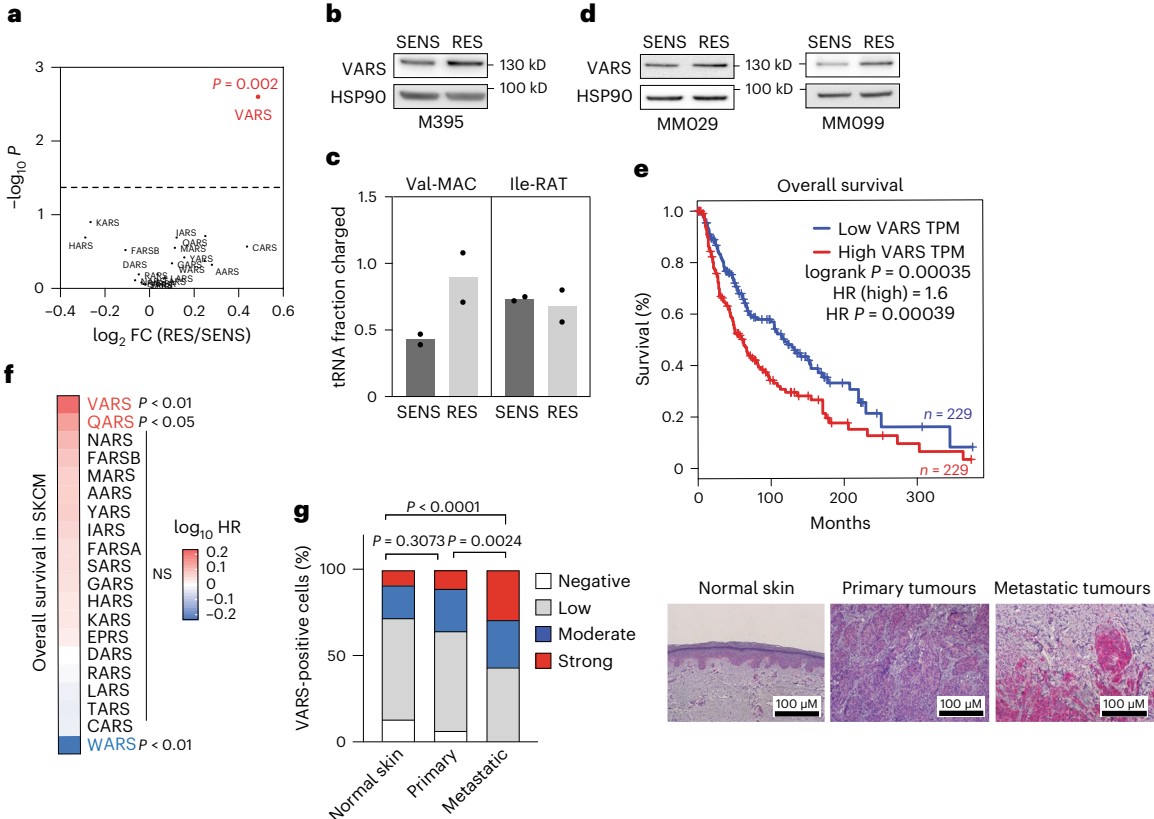

**Fig. 3 | VARS activity is increased in melanoma resistant to targeted therapy.** **a**, Expression of aaRSs in M395 RES cultures (proteomics; $n$ = 3, two-tailed unpaired $t$-test). **b**, VARS expression in SENS and RES cultures as confirmed by western blot ($n$ = 2). **c**, Val-MAC tRNA and Ile-RAT tRNA aminoacylation analysis of M395 SENS and RES cells (mean ± s.e.m. of $n$ = 2 independent replicates). **d**, VARS expression in SENS and RES to dual MAPK-targeted therapy ($n$ = 2). **e**, Survival analysis of patients with SKCM with high or low expression of VARS. TPM, transcripts per million. HR, hazards ratio. **f**, Evaluation of the skin of patients with cutaneous melanoma overall survival (data from TCGA) related to the expression of all the tRNA synthetase. VARS expression is associated with the poorest prognosis in patients with SKCM. The $P$ values were obtained from GEPIA2. **g**, VARS immunostaining in patients with normal skin ($n$ = 12) and primary ($n$ = 12) or metastatic ($n$ = 21) melanoma biopsies. The representative images and quantifications are shown, and a chi-squared test was performed.

in RES cultures (Fig. 2d and Extended Data Fig. 2b). Of note, the GTG codon is, among the four valine codons, the most represented in the human transcriptome, as evidenced by the analysis of distribution of valine codons in the human transcriptome (Extended Data Fig. 2c).

Strikingly, tRNA-Val-CAC upregulation was confirmed by tRNA quantitative polymerase chain reaction with reverse transcription (RT–qPCR) (after tRNA demodification) in three patient melanoma cultures (Fig. 2e) and by northern blot in SENS and RES MM029 cultures (Fig. 2f and Extended Data Fig. 2d). tRNA-Val-CAC was consistently upregulated in all patient-derived RES lines as compared with their isogenic SENS counterparts. tRNA-Cys-GCA was downregulated in two out of the three tested lines. tRNA-Ser-AGA and tRNA-Ser-TGA did not change (Fig. 2e). Taken together, these findings highlight the importance of valine and its cognate tRNAs in the proteome rewiring of RES melanoma.

**VARS activity is upregulated in therapy-resistant melanoma**

Interestingly, the analysis of tRNA aminoacyl transferases uncovered a significant and specific upregulation of VARS, the valine tRNA aminoacyl transferase, in RES melanoma cultures (Fig. 3a and Extended Data Fig. 3a). This was consistent with the increase in tRNA-Val and the enrichment of valine in proteins upregulated in RES cultures (Figs. 1c and 2a). Strikingly, VARS protein expression was enhanced in RES M395 melanoma cultures, as compared with its SENS counterpart, and correlated with an increase in VARS aminoacylation activity (the aminocylation of isoleucine-RAT tRNA remains unchanged; Fig. 3b,c and Extended Data Fig. 3b). VARS protein was also upregulated in both RES MM029 and

MM099 melanoma cultures as compared with their SENS counterparts (Fig. 3d and Extended Data Fig. 3b). Importantly, in The Cancer Genome Atlas (TCGA) database taken from GEPIA[26], VARS is highly expressed in skin cutaneous melanoma (SKCM) and significantly upregulated in tumours as compared with normal melanocytes (Extended Data Fig. 3c,d). There, high VARS expression is associated with lower overall survival in patients with melanoma (Fig. 3e). Hence, within the aaRS family, VARS expression was the most significantly associated with poor overall survival in SKCM (Fig. 3f). Finally, immunohistochemical analyses revealed that VARS protein levels are increased in human biopsies of metastatic melanoma as compared with primary tumour or normal skin (Fig. 3g).

**VARS depletion sensitizes melanoma cells to MAPK therapy**

To assess the functional contribution of VARS in melanoma survival and resistance, we knocked down VARS in various melanoma cells. VARS depletion correlated with a consistent decrease in the aminoacylation of tRNA-Val-MAC in RES melanoma cultures but did not affect the aminoacylation of tRNA-Ile-RAT (Fig. 4a). Strikingly, VARS depletion significantly re-sensitized RES melanoma cells to BRAF inhibition (vemurafenib) in a dose-dependent manner. Surprisingly, VARS depletion did not significantly affect melanoma cells survival in untreated conditions (Fig. 4b,c). Distinct drug-tolerant states have been identified in melanoma[27], highlighting the ability of melanoma cells to escape the drug combination through distinct mechanisms. We depleted VARS in patient resistant cultures harbouring distinct

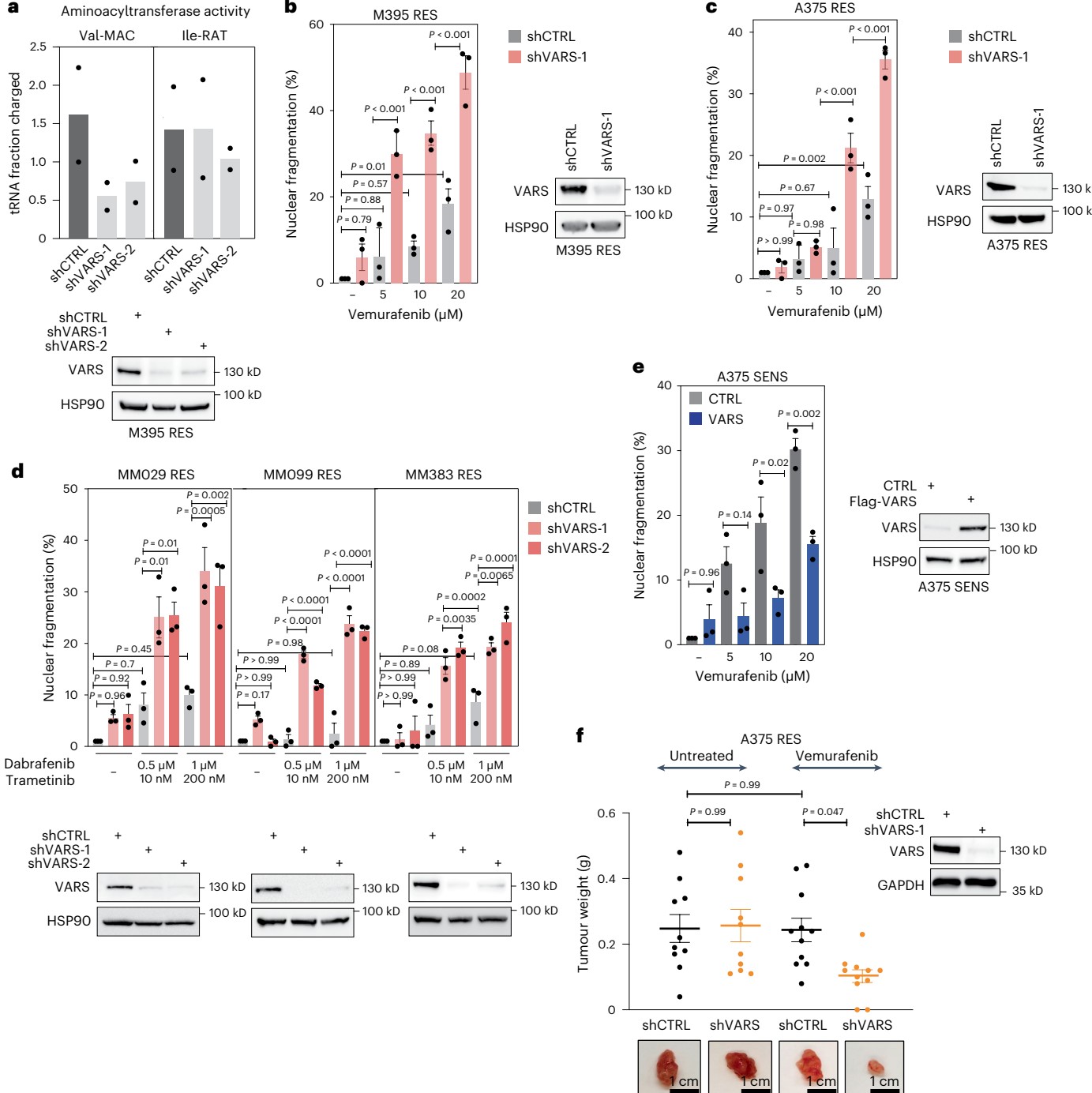

**Fig. 4 | VARS depletion re-sensitizes resistant melanoma cells to MAPK-based therapy. a**, Val-MAC-tRNA and Ile-RAT-tRNA aminoacylation activity of M395 RES cells depleted of VARS using two shRNAs (mean ± s.e.m. of $n = 2$ independent replicates). **b–d**, Cell death of patient-derived melanoma measured by fluorescence staining with PI by FACS of M395 RES (**b**), A375 RES (**c**) and a series of patient-derived melanoma cultures resistant to the drug combination BRAFi/MEKi (**d**). The cells were treated or not treated with vemurafenib (as indicated in **b** and **c**) or with dabrafenib/trametinib (as indicated in **d**) and depleted or not

of VARS. **e**, As in **b** and **c** but with SENS cells overexpressing VARS. CTRL, control overexpression. **f**, A375 RES control or VARS-depleted melanoma cells were xenografted in mice and treated or not with vemurafenib (25 mg kg⁻¹) ($n = 10$ for untreated and $n = 11$ for vemurafenib treated mice). Tumour weight mean ± s.e.m. was assessed and plotted. The mean ± s.e.m. of $n = 3$ independent replicates is indicated for **b**, **c**, **d** and **e**. A two-way analysis of variance was performed for **b–f** ($P < 0.0001$). Tukey's multiple comparisons are indicated in the figures.

drug tolerant phenotypes (here, cells are resistant to the combination BRAFi/MEKi—trametinib and dabrafenib: MM029 and MM099 in the invasive state and MM383 in the neural crest stem cells state[27]). VARS depletion invariably and significantly re-sensitized these melanoma cultures (Fig. 4d). Here, again, VARS depletion did not significantly impact the survival of melanoma cells in untreated conditions.

Together, these results indicate that VARS contributes to the resistance of melanoma cells to MAPK therapy.

Strikingly, enhanced VARS expression in SENS A375 cells was sufficient to protect them from vemurafenib, indicating that VARS expression promotes tolerance to BRAF inhibition (Fig. 4e). Finally, control or VARS-depleted RES A375 cells were implanted in NOD-SCID mice

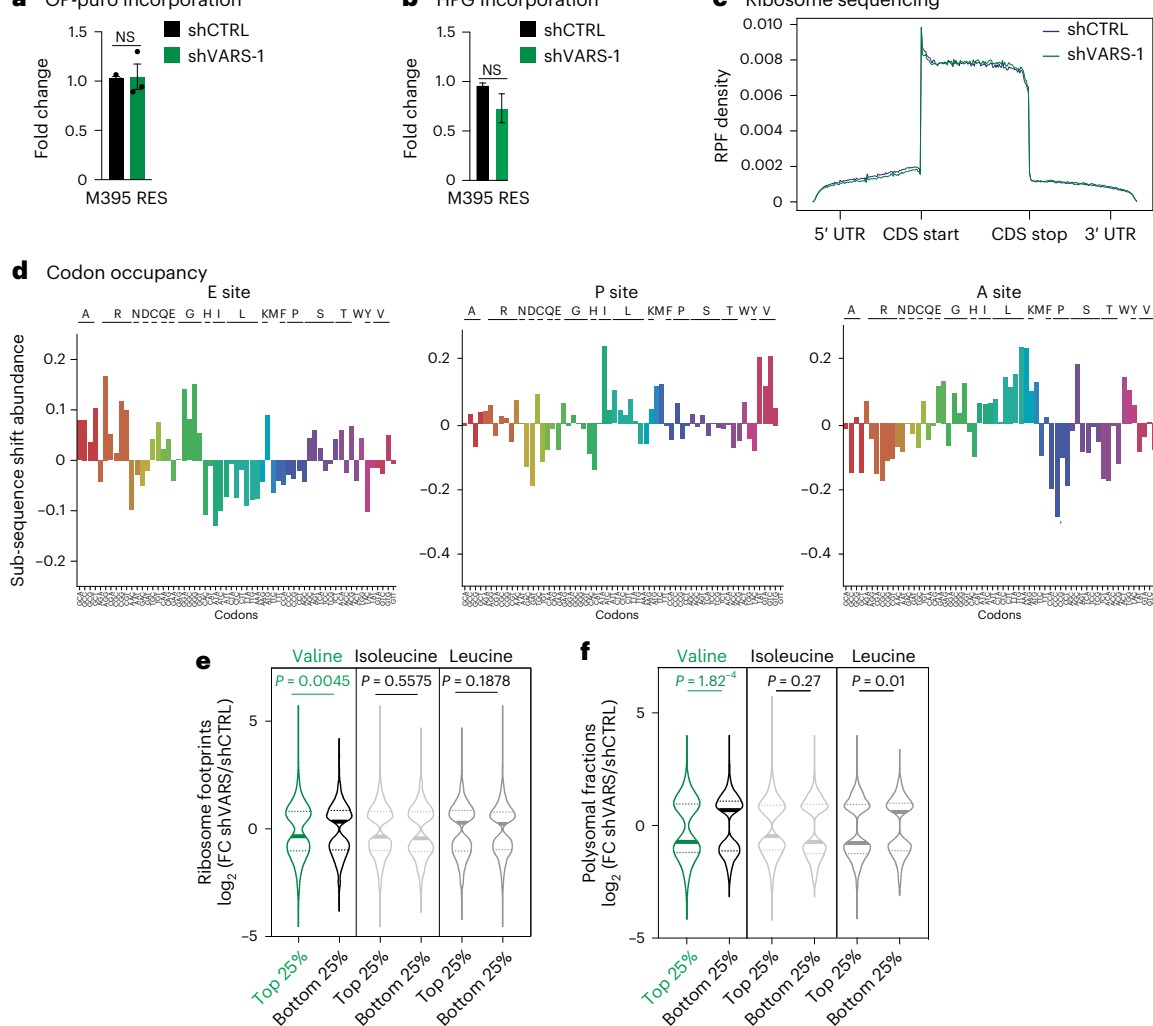

**Fig. 5 | VARS regulates the translation of valine rich transcripts. a,b**, OP-puro (**a**) and HPG (**b**) incorporation relative to control in M395 RES melanoma cultures depleted or not depleted of VARS (fold mean ± s.e.m. of $n$ = 3 independent replicates; a two-tailed unpaired $t$-test was performed). **c**, Metagene density profiles depicting global RPFs in coding sequences (CDS) in M395 RES cells after VARS depletion (green) as compared with control (blue). The $y$ axis shows the average intra-gene normalized density of RPFs for each transcript. UTR, untranslated region. **d**, Diricore analysis plots at the E site, P site and A site (positions 9, 12 and 15, respectively), analysing differential RPFs density at codons in M395 RES shVARS as compared with shCTRL. **e,f**, Violin plots representing the $\log_2$ FC of significantly regulated transcripts of shVARS as compared to control in ribosome sequencing (**e**) or polysome sequencing (**f**). The transcripts are sorted based on their enrichment (top 25%) or impoverishment (bottom 25%) in the indicated amino acids (two-tailed unpaired $t$-test).

and treated or not with vemurafenib. While VARS depletion did not impact tumour growth in untreated mice, it strongly synergized with vemurafenib treatment to prevent growth of resistant tumours (Fig. 4f and Extended Data Fig. 4a). Therefore, VARS depletion synergizes with MAPK therapy to limit melanoma survival in vitro and to limit resistant melanoma tumour growth upon treatment in vivo.

**VARS regulates mRNA translation of valine-enriched mRNAs**

Next, we assessed the impact of VARS depletion on global mRNA translation and protein synthesis by OP-Puro and L-homopropargylglycine (HPG) incorporation in RES M395 cells (Fig. 5a,b and Extended Data Fig. 5a). Surprisingly, VARS depletion did not significantly impact global mRNA translation. Hence, VARS depletion was not associated with any sign of integrated stress response in M395 and A375 cells (Extended Data Fig. 5b). Moreover, we performed ribosome profiling experiments with RES M395 cultures depleted or not of VARS. Again, VARS depletion was not associated with global changes in ribosome-protected fragment (RPF) density across mRNAs (Fig. 5c). Of note, we did not detect any specific stalling of ribosomes at translation initiation or termination

sites (Fig. 5c). Furthermore, no consistent change in codon occupancy at any of the ribosome sites was highlighted in VARS-depleted cultures (that is, Diricore analysis[28] for the E-site, P-site and A-site; Fig. 5d). Taken together, our results indicate that VARS depletion does not impact mRNA translation in a global manner in resistant melanoma cells.

To assess whether VARS depletion may affect the translation of specific mRNAs, we further classified transcripts based on their enrichment in valine codons (top 25% and bottom 25%). Strikingly, we found that VARS depletion led to a significant and specific reduction of ribosome occupancy at valine-enriched transcripts compared with those impoverished in valine codons (Fig. 5e). Importantly, ribosome occupancy remained unchanged at transcripts enriched for leucine or isoleucine codons (branched chain amino acids) upon VARS depletion, pointing towards a specific effect of VARS depletion on the translation of valine-enriched transcripts (Fig. 5e). To validate our findings, we performed a polysome profiling experiment in control and VARS-depleted RES M395 cells. Here, again, VARS depletion did not significantly impact the polysome profiles (Extended Data Fig. 5c). However, consistent with our previous observations, we found that VARS depletion significantly

affected the translation of transcripts enriched in valine codons and had no or little effect on transcripts enriched in isoleucine or leucine codons (Fig. 5f). Together, these data indicate that VARS depletion leads to specific translation defects at transcripts enriched in valine codons in resistant melanoma.

### Identification of VARS potential translational targets

To relate the translation defects observed at valine-enriched transcripts to protein levels, we performed quantitative proteomics using control and VARS-depleted RES M395 cells. First, proteins downregulated upon VARS depletion were enriched in valine (Fig. 6a), while no enrichment in valine was found in proteins upregulated in RES cultures (Extended Data Fig. 6a,b and Supplementary Table 3). Of note, none of the tRNA aminoacyl transferases was found to be upregulated upon VARS depletion (Extended Data Fig. 6c). Next, we integrated the proteomics data with transcriptomics data of VARS-depleted cultures (Fig. 6b and Extended Data Fig. 6d). Of note, VARS depletion had little impact on global mRNA expression. We defined the VARS translational signature ($n = 84$) as candidates (1) downregulated at protein levels, (2) whose mRNA is poorly impacted by VARS depletion and (3) enriched in valine. Hence, chi-square analyses showed that VARS signature is predictive of translationally downregulated genes in both ribosome and polysome sequencing (Fig. 6c,d). Interestingly, the VARS signature is enriched in pathways such as FA metabolism and mTORC1 signalling, among others (Fig. 6e), which play a key role in melanoma resistance to MAPK therapy[22,29,30]. We next sought to identify VARS targets whose regulation functionally impacts melanoma resistance from the established VARS signature (Fig. 6f). Interestingly, this analysis highlighted 22 candidates, which clustered in hallmarks of metabolic processes (Fig. 6g). In particular, five proteins were linked to mTORC1 signalling (that is, QDPR, SLC7A5, PSAT1, FDXR and TXNRD1), four proteins to adipogenesis (that is, QDPR, hydroxyacyl-CoA dehydrogenase (HADH), DBT and BCL2L13), three proteins to the unfolded protein response (that is, SLC7A5, PSAT1 and ALDH18A1) and three proteins to FA metabolism (that is, HADH, HMGCL and CRYZ) (Fig. 6g). To assess the functional importance of these candidates in melanoma resistance to MAPK therapy, we performed a systematic functional esiRNA screen (Fig. 6h). RES MM029 melanoma cultures (resistant to the combination dabrafenib/trametinib) (Fig. 4d) were transfected with control or target-specific esiRNA ($n = 22$) and were treated or not with MAPK therapy (the combination dabrafenib/trametinib). Cellular sensitivity to the drug combination was measured by propidium iodide (PI) staining. We selected the five top candidates (HADH, KIF13B, SLC7A5, QDPR and GOLT1B) whose depletion induced high cell death upon treatment but not in untreated conditions (Fig. 6h). Strikingly, VARS depletion affected the translation of all selected candidates, as highlighted in both polysome and ribosome profiling experiments (Fig. 6i,j and Extended Data Fig. 6e,f). Consistently, VARS depletion in resistant melanoma cultures led to a systematic decrease in the expression of the five identified target proteins (Fig. 6k). These data demonstrate

that VARS controls the mRNA translation of valine-enriched proteins which promote resistance of melanoma to MAPK therapy.

### VARS promotes FA oxidation through HADH translation

The patients with $BRAF^{V600E}$ melanoma from TCGA were clustered according to the previously described VARS signature and analysed with single-sample gene set enrichment analysis (ssGSEA). The patients with VARS[high] (that is, patients with the highest score for VARS signature; see Methods), were significantly enriched in FA metabolism (Fig. 7a and Extended Data Fig. 7a). FA oxidation promotes resistance of melanoma cells to MAPK therapy[22,29,30]. Accordingly, FA oxidation activity was significantly enhanced in RES M395 cultures (Fig. 7b). Etomoxir treatment, which inhibits FA oxidation, was sufficient to re-sensitize RES M395 melanoma cultures to BRAFi, highlighting the importance of FA oxidation in RES to MAPK therapy (Fig. 7c and Extended Data Fig. 7b). Since HADH (previously identified, Fig. 6h) serves an important role in the FA oxidation, we assessed its expression in RES M395 cultures. HADH protein levels were enhanced in RES cultures and correlated with an increase in VARS expression (Fig. 7d). HADH expression also correlated with VARS expression in different normal skin and melanoma biopsies (Extended Data Fig. 7c). Therefore, we surmised that VARS could promote melanoma resistance at least through regulating HADH mRNA translation and FA oxidation. HADH protein expression correlated with VARS expression in various melanoma lines: HADH was decreased upon VARS depletion in RES melanoma cells (Fig. 7e and Extended Data Fig. 7d) and it was increased upon VARS upregulation in SENS melanoma cells (Fig. 7f). The importance of VARS in the regulation of HADH mRNA translation was confirmed by ribosome immunoprecipitation using RPL22-Flag expressing melanoma cells (as in ref. 31). There, VARS depletion led to a decrease in ribosome content at HADH transcripts (Extended Data Fig. 7e). Accordingly, VARS depletion was sufficient to reduce FA oxidation activity in RES M395 cultures (Fig. 7g), while VARS overexpression significantly promoted FA oxidation activity in SENS M395 cultures (Fig. 7h). Consistent with the role of HADH in FA oxidation, its depletion in two RES melanoma cultures compromised cellular FA oxidation activity and efficiently re-sensitized RES M395 to BRAFi (Fig. 7i,j and Extended Data Fig. 7f,g). These data show that VARS regulates FA oxidation in resistant melanoma cultures, at least through the regulation of HADH mRNA translation.

### Discussion

Here, we found that VARS plays a key role in driving therapy resistance in melanoma through promoting selective mRNA translation of valine-enriched transcripts. We identified several candidates whose mRNA translation relies on VARS and whose depletion sensitizes resistant melanoma to MAPK therapy. Among them, VARS regulates the translation of $HADH$, a valine-enriched protein involved in FA oxidation, whose activity supports adaptation of $BRAF^{V600E}$ melanoma to MAPK therapy[22,29,30]. VARS depletion affects HADH levels, reduces cellular FA oxidation and re-sensitizes resistant melanoma cells to MAPK therapy.

---

**Fig. 6 | VARS translation targets in the resistant melanoma phenotype.**
**a**, A heat map representing amino acid enrichment analysis of proteins (proteomics) downregulated in M395 RES depleted of VARS (chi-squared test). **b**, A graph representing genes commonly detected by RNA sequencing ($y$ axis) and proteomics ($x$ axis) of M395 RES VARS-depleted cells (shVARS-1) compared with control cells (shCTRL). Four groups were assigned depending on their $\log_2$ FC. 'Translation down' and 'translation up' groups were appointed with: $-1 < \log_2$ FC ($y$ axis) $< 1$ and $\log_2$ FC ($x$ axis) $< -0.32$ or $1 < \log_2$ FC ($x$ axis) $< 1$ and $\log_2$ FC ($x$ axis) $> 0.32$, respectively. **c,d**, The VARS signature was superposed with genes presenting differential RPFs (**c**) or polysomal fractions (**d**) of control (shCTRL) or VARS-depleted cells (shVARS) (chi-squared test). **e**, A bubble plot of the top five enriched terms (using Gene Ontology (GO) Hallmark) for VARS signature ($n = 84$). The size of the bubble represents the $-\log_{10}$ (FDR $q$ value). **f**, A Venn diagram highlighting 22 proteins commonly overlapping in proteins

that are upregulated during RES ($n = 706$) and in VARS signature ($n = 84$). **g**, A bubble plot of the top five enriched (using GO Hallmark) for the 22 valine-enriched candidates. The size of the bubble represents the $\log_{10}$ (FDR $q$ value). **h**, Cell death measurement of MM029 resistant to trametinib/dabrafenib by fluorescence staining with PI by FACS after the depletion of the indicated candidate using specific esiRNA and after treatment or not with the indicated drug combination. The red arrows depict the top five candidates (showing highest percentage of nuclear fragmentation upon treatment with no or little nuclear fragmentation in untreated condition) ($n = 1$). **i,j**, Heat maps highlighting the $\log_2$ (FC shVARS/shCTRL) of the top five candidates (HADH, KIF13B, SLC7A5, QDPR and GOLT1B) in ribosome footprints (**i**) and polysomal fractions (**j**). The red squares represent significant values. **k**, Western blot analysis of VARS and the indicated proteins expression in control (shCTRL) and VARS (shVARS)-depleted M395 RES cultures ($n = 1$).

We found that the proteome of drug-resistant melanoma cells is biased towards the use of valine, among other amino acids. Hence, valine tRNA isoacceptors are upregulated in resistant melanoma.

In particular, tRNA-Val-CAC, decoding GTG codon, the most represented valine codon in human genes, is systematically upregulated in drug resistant melanoma. This suggests that a specific mRNA

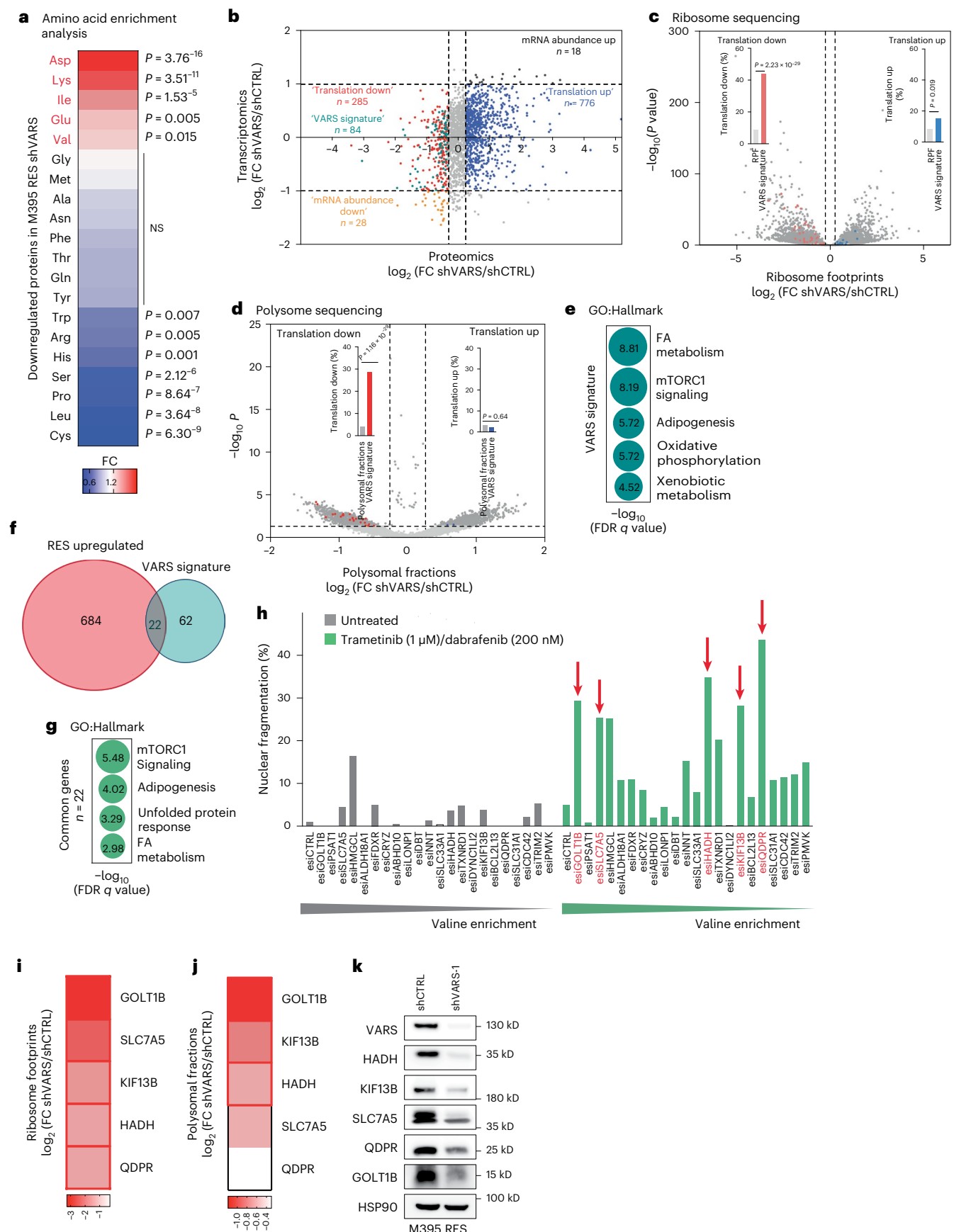

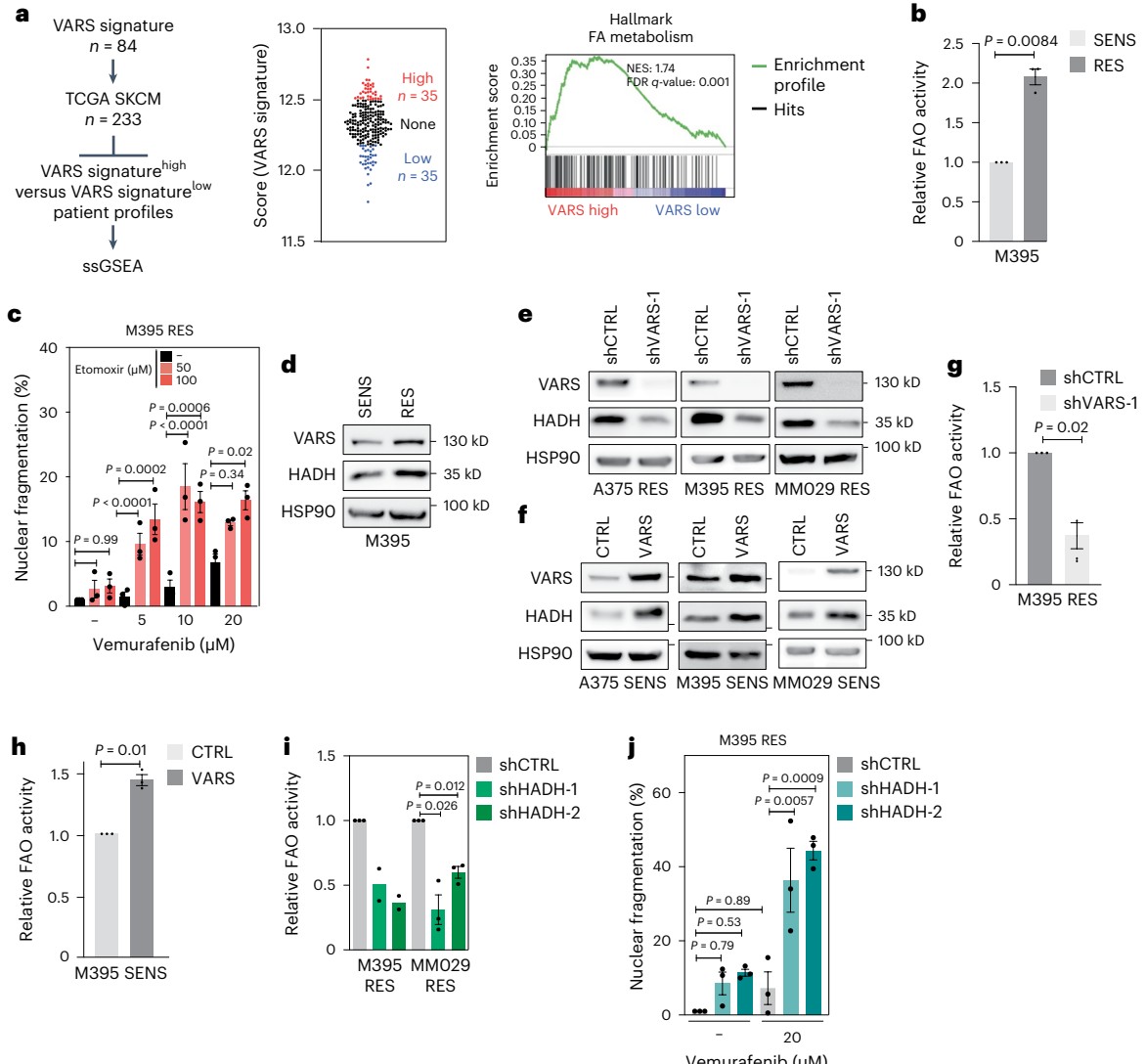

**Fig. 7 | VARS controls the FA metabolism in resistant melanoma cells.**
**a**, Experimental set-up showing the score attribution for VARS signature (high and low) in patients with *BRAF^V600E*-mutated SKCM melanoma from TCGA (*n* = 233). GSEA analysis enrichment of the FA metabolism in VARS high versus VARS low TCGA melanomas. **b**, FA oxidation (FAO) activity measured in SENS and RES M395 patient-derived cultures (mean fold change ± s.e.m.; *n* = 3 independent biological replicates, two-tailed unpaired *t*-test). **c**, Cell death measurement by fluorescence staining with PI by FACS in M395 RES cells treated or not with etomoxir and vemurafenib at indicated concentrations (mean fold change ± s.e.m.; *n* = 3 independent biological replicates; two-way analysis of variance (*P* < 0.0001)). Tukey's multiple comparisons test are indicated. **d**–**f**, Western blot of VARS and HADH protein expression in SENS and RES M395 cells

(**d**), in A375, M395 and MM029 RES cells depleted of VARS (**e**), and in A375, M395 and MM029 SENS cells overexpressing (VARS) or not (control) VARS (**f**). **g**,**h**, FAO activity measured in M395 RES cells depleted or not for VARS (**g**) and in M395 SENS cells overexpressing (VARS) or not (control) (**h**) (fold mean ± s.e.m. of *n* = 3 independent replicates, two-tailed unpaired *t*-test). **i**, FAO activity measured in M395 and MM029 RES cells depleted of HADH (shHADH-1 and shHADH-2) compared with control cells (shCTRL) (mean fold change ± s.e.m.; *n* = 2 M395 RES and *n* = 3 MM029 RES, two-tailed unpaired *t*-test). **j**, Cell death measurement by fluorescence staining with PI by FACS in control (shCTRL) or HADH (shHADH) depleted M395 RES cells treated or not with vemurafenib (as indicated) (mean fold change ± s.e.m.; *n* = 3 independent biological replicates, two-way analysis of variance (*P* < 0.0001). Tukey's multiple comparison tests are indicated.

translation programme using valine is prominent in resistant melanoma. Importantly, valine tRNA biogenesis was shown to promote tumorigenesis and survival in leukaemia. There, valine-restricted condition reduced translation rates of specific mRNAs and compromised T-cell acute lymphoblastic leukemia growth[32]. Our data showed that VARS knockdown does not significantly compromise melanoma cellular survival nor tumour growth in mice, unless cells or tumours are treated with MAPK therapy. As such, VARS activity protects melanoma from MAPK therapy, at least through promoting *HADH* translation and FA oxidation cellular activity. Importantly, VARS depletion does not completely abolish the cellular valine aminoacylation activity. The remaining VARS activity may maintain melanoma survival in untreated conditions. Consistently, we do not observe any global

changes in mRNA regulation upon VARS depletion. Other aaRSs may compensate the limited VARS activity. Although aaRSs are highly conserved enzymes[33], mistranslation can occur with amino acids sharing similar structure, leading to generation of nascent proteins with mis-incorporated amino acids. For example, isoleucine aaRSs can also accommodate valine to tRNA^Ile, although at a rate of 1/200 (refs. 34,35). Similarly, VARS can also load threonine to tRNA^Val with lower affinity[36]. Studies on cross-aminoacylation showed that mitochondrial aminoacyl transferases may be active at corresponding cytoplasmic tRNAs[37,38]. However, these hypotheses remain to be tested. Of note, VARS knockdown in melanoma cells is not associated with any change in expression of other aaRSs. Another intriguing possibility is that specific codon reassignment may take place upon VARS knockdown. This

was previously described as the 'codon capture theory'[39]. Recently, tryptophan aaRSs was shown to drive tryptophan-to-phenylalanine codon reassignment in cells deprived of tryptophan[40]. These hypotheses, which may explain how cells compensate the lack of VARS activity, will be further studied.

VARS expression and activity are enhanced in resistant melanoma cultures and correlate with upregulation of valine tRNAs isoacceptors. We could not highlight any direct regulation of VARS by BRAF or MEK inhibition, but we cannot exclude that VARS could be downstream of the PI3K pathway, prominent in resistant melanoma[41]. A recent study demonstrates that NOTCH transcriptionally regulates VARS expression[32]. This remains to be tested in our model. Also, VARS could auto-activate its own transcription in melanoma, as does alanine aaRSs by direct binding to DNA[42]. Finally, VARS expression can be controlled by the levels of its isoacceptor tRNAs or of their aminoacylation. Indeed, low levels of uncharged tRNAs could promote the expression and activation of specific aaRS to re-establish normally charged tRNAs[43]. Understanding the mechanisms leading to the upregulation of VARS or other aaRS in human diseases is an area of future investigation. Finally, we found that VARS does not impact BRAF expression nor MAPK pathway activation in the tested $BRAF^{V600E}$ melanoma SENS or RES cultures (Extended Data Fig. 7h,i).

The possibility that VARS may promote resistance through a non-tRNA dependent mechanism cannot be fully excluded. However, our study shows that VARS regulates the translation of specific mRNAs encoding valine-enriched proteins. Our functional esiRNA screen identified five candidate translation targets of VARS (that is, HADH, KIF13B, SLC7A5, QDPR and GOLT1B), whose depletion re-sensitizes resistant melanoma cells to MAPK therapy. This may suggest that VARS integrates convergent mechanisms of therapy resistance in melanoma. Interestingly, our results link VARS activity to the maintenance of active FA oxidation in resistant melanoma. Several studies connected aberrant FA oxidation activity to therapeutic resistance in melanoma[22,30] and other cancers[44,45]. For instance, inhibition of FA oxidation enhanced the sensitivity of resistant breast cancer to radiotherapy[46]. Here, we show that inhibition of FA oxidation (by HADH knockdown or etomoxir treatment) efficiently re-sensitized resistant melanoma cells to MAPK therapy, similarly to VARS knockdown. It remains to be tested whether VARS activity is essential in other malignancies, which rely on high FA oxidation activity.

Recent studies have revealed the unexpected importance of aaRSs in human diseases, such as cancers[47–49]. For instance, methionine aaRS is overexpressed in non-small cell lung cancer and is associated with poor clinical outcome[50]. Threonine aaRS is upregulated in human ovarian cancer and correlates with angiogenic tumour progression[51]. Tryptophan aaRS is involved in cancer metastasis, angiogenesis and neuronal and autoimmune diseases in humans[52]. Leucine aaRS is a tumour suppressor in breast cancer[10]. Together, these findings encourage additional studies on understanding the role of aaRSs in human diseases and unveil the potential of aaRSs as therapeutic targets.

Taken together, we highlighted VARS as a driver of resistance to targeted therapy in melanoma (see model, Extended Data Fig. 8) and uncovered attractive therapeutic opportunities for patients with RES melanoma.

## Online content

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

[1]Laboratory of Cancer Signaling, GIGA Institute, University of Liège, Liège, Belgium. [2]Laboratory of Cancer Biology, GIGA Institute, University of Liège, Liège, Belgium. [3]Division of Oncogenomics, Oncode Institute, The Netherlands Cancer Institute, Amsterdam, The Netherlands. [4]Laboratory of Cancer Stemness, GIGA Institute, University of Liège, Liège, Belgium. [5]Laboratory of Experimental Pathology, GIGA Institute, University of Liège, Liège, Belgium. [6]Bioinformatics platform, GIGA Institute, University of Liège, Liège, Belgium. [7]Laboratory of Molecular Regulation of Neurogenesis, GIGA Institute, University of Liège, Liège, Belgium. [8]Center for Interdisciplinary Research on Medicines—Laboratory of Medicinal Chemistry, University of Liège, Liège, Belgium. [9]Department of Hematopoietic Biology and Malignancy, MD Anderson Cancer Center, Houston, TX, USA. [10]UMR 7156 – Molecular Genetics, Genomics, Microbiology, University of Strasbourg/CNRS, Strasbourg, France. [11]WELBIO department, WEL Research Institute, Wavre, Belgium. [12]Laboratory for Molecular Cancer Biology, Department of Oncology, KU Leuven, Leuven, Belgium. [13]Center for Cancer Biology, VIB, Leuven, Belgium. [14]Erasmus MC, Department of Genetics, Rotterdam University, Rotterdam, The Netherlands. [15]These authors contributed equally: Marine Leclercq, Miguel Susaeta Ruiz. ✉e-mail: Pierre.Close@uliege.be

## Methods

Our research complies with all relevant ethical regulations. Our in vivo experiments are approved by the Ethical Committee of the University of Liege.

### Cell culture

A375 melanoma cell lines were purchased from the American Type Culture Collection (CRL-1619) and Lenti-X293 T from Sigma-Aldrich and were grown in Dulbecco's modified Eagle medium supplemented with 10% of fetal bovine serum, 1% glutamine and 1% penicillin–streptomycin in a humidified atmosphere containing 5% $CO_2$ at 37 °C. A375 vemurafenib-resistant cells were generated by increasing doses of vemurafenib up to 1 µM. M395 (SENS and 1 µM vemurafenib RES) lines were from the laboratory of R. Lo (UCLA Division of Dermatology) and were grown in RPMI medium with the supplementations mentioned above. MM029, MM099 and MM383 naive and their resistant counterparts were from J.C. Marine (KU Leuven). These cells were grown in Ham's F-10 medium supplemented with 10% fetal bovine serum, 1% glutamine and 1% penicillin–streptomycin. MM074 cells were from G. Ghanem (Institut J. Bordet, Université Libre de Bruxelles). The MM074 line was grown in RPMI Ham's F-10 medium with supplementation.

### Antibodies, esiRNA, shRNA and other reagents

The antibodies, compounds, endoribonuclease-prepared small interfering RNAs (esiRNAs) and short hairpin RNA (shRNA) sequences, as well as tRNA-specific qPCR primers, used in this study are listed in Supplementary Table 4.

### tRNA sequencing

Total RNA was extracted using TriPure isolation method (11667157001, Roche). After validating the integrity and purity of each RNA, tRNAs were isolated from the total RNA resolved on urea–polyacrylamide gels in the size range of 60–100 nt. Purified tRNAs were then demethylated for $m^1A$ and $m^3C$ and hydrolysed by limited alkaline hydrolysation with carbonate buffer. Partially hydrolysed fragments were then dephosphorylated by calf intestinal phosphatase and rephoshorylated at 5′ by T4 polynucleotide kinase. For the library preparation, tRNA fragments that were partially hydrolysed and re-phosphorylated were converted to complementary DNA using the Small RNA Library Prep Set kit from Illumina. The libraries were qualified and absolutely quantified using Agilent BioAnalyzer 2100. tRNA sequencing was performed on an Illumina NextSeq 500 by single-read sequencing at 75 bp. tRNA alloisoacceptors were quantified by the addition of all isodecoder genes for tRNA sharing the same amino acid. The tRNA isoacceptors were quantified by the addition of all significant isodecoder genes for each tRNA sharing the same anticodon.

### Northern blot

A total of 1.5 g of trizol extracted RNAs from SENS and RES MM029 melanoma cultures were subjected to electrophoresis in 8 M urea 8% polyacrylamide gel (19:1) in 1× Tris–borate–EDTA buffer. The RNAs were then transferred on a positively charged nylon membrane at 0.2 A for 1 h in 0.5× Tris–borate–EDTA buffer using the Trans-Blot SD Semi-Dry system (Bio-Rad). The RNAs were crosslinked using a Spectrolinker ultraviolet crosslinker (Invitrogen). Hybridization was performed using the DIG Easy Hyb solution (Roche) at 42 °C. The probes were labelled using the DIG Oligonucleotide 3′-End Labeling kit, second generation, and hybridized overnight at 42 °C. The post-hybridization washes included two washes with 2× saline-sodium citrate (SSC), 0.1% sodium dodecyl-sulfate (SDS) for 5 min at room temperature followed by two washes with 0.5× SSC, 0.1% SDS for 15 min at 42 °C. Detection was achieved using the DIG Luminescent Detection kit (Roche) according to the manufacture's protocol. All probe sequences are listed in the Supplementary Table 4.

### Proteomic analysis

SENS or RES M395 cells and RES M395 cells depleted or not of VARS were lysed in a 2% SDS lysis buffer (50 mM Tris–HCl pH 8.0, 150 mM sodium chloride (NaCl), 10 mM sodium fluoride (NaF), 1 mM trisodium phosphate ($Na_3PO_4$), complete proteinase inhibitor (Roche) and phosphatase inhibitor (phosphoSTOP, Roche). All the samples were performed in triplicates. The protein extracts were tricarboxylic acid precipitated overnight in 20% trifluoroacetic acid (TFA) in 90% acetone at −20 °C. After washes and centrifugation, pellets were then resuspended in 5 M urea, 50 mM Tris–HCl, pH 8.0, Complete proteinase inhibitor (Roche) and quantified by Qubit fluorometry (Life technologies). A 50 µg aliquot of each sample was diluted with 25 mM ammonium bicarbonate and reduced with 10 mM dithiothreitol followed by alkylation with 15 mM iodoacetamide at room temperature. The proteins were then digested with 2.5 µg trypsin (Promega) at 37 °C for 18 h and quenched with formic acid and peptides cleaned by solid-phase extraction using the Empore C18 plate (3 M). A total of 2 µg per sample was analysed by nano liquid chromatography tandem mass spectrometry (MS/MS) with a Waters M-Class high-performance liquid chromatography system interfaced to a Thermo Fisher Fusion Lumos. The peptides were loaded on a trapping column and eluted over a 75 µm analytical column at 350 nl min$^{-1}$; both columns were packed with Luna C18 resin (Phenomenex) using a 4 h gradient. The mass spectrometer was operated in data-dependent mode, with MS and MS/MS performed in the Orbitrap at 60,000 full width at half maximum resolution and 15,000 full width at half maximum resolution, respectively. Advanced peak determination was turned on. The instrument was run with a 3 s cycle for MS and MS/MS. The data were processed through the MaxQuant software v1.6.0.16 (www.maxquant.org), normalized using LFQ algorithm and searched using Andromeda with the Swissprot Human database.

### RNA sequencing

Total RNA was extracted using the TriPure isolation method (11667157001, Roche). Quality and integrity of RNA was assessed before proceeding to sequencing. The mRNA library preparation was performed using the TruSeq Stranded mRNA kit (20020595, Illumina). Sequencing was performed using the NovaSeq S1 (100 cycles) standard workflow.

### Codon and amino acid enrichment analysis

Codon and amino acid enrichment analysis were assessed as followed for both proteomics and the GSE50509 dataset. Codon enrichment was evaluated using a chi-squared test, comparing the codon distribution of each of the genes from each group to the human transcriptome. The codon frequency of the longest transcript corresponding to each gene was used for the analysis. A transcript was considered enriched for a specific codon if its content was found in the top 25% of this codon's distribution in the transcriptome. The same analysis was also applied for the amino acid enrichment analysis.

### tRNA qPCR

Total RNAs were extracted using using the TriPure isolation method (11667157001, Roche). At least 3 µg of total RNAs were used for RNA demethylation followed by RNA reprecipitation and first-strand cDNA synthesis (rtStar tRNA Pretreatment and First-Strand cDNA Synthesis kit, AS-FS-004, Arraystar). These steps were performed according to the manufacturer's recommendations. Quantitative evaluation of specific tRNA was determined by qPCR using SYBR Green and pre-designed tRNA primer sets (Arraystar) in a LightCycler 480 System (Roche). Briefly, qPCR was run with a first step at 95 °C for 10 min (one cycle) followed by an amplification step at 95 °C for 10 s then at 60 °C for 1 min (40 cycles) and a melting curve analysis. The primers are listed in Supplementary Table 4.

## Polysome profiling

SENS or RES M395 cells, and M395-resistant cells depleted or not of VARS were gently scraped in phosephate-buffered saline (PBS) containing 100 µg ml⁻¹ cycloheximide (CHX) on ice. After centrifugation at 200$g$ for 5 min at 4 °C, the cells were lysed in hypotonic buffer containing 2.5 mM $MgCl_2$, 5 mM TRIS pH 7.5, 1.5 mM KCl and 1× protease inhibitors. Subsequently, CHX, dithiothreitol, RNAse inhibitor and triton X-100 followed by sodium deoxycholate were sequentially added to solubilize cytosolic and endoplasmic reticulum-associated ribosomes as in ref. 53. The lysates were then centrifuged at 16,000$g$ for 7 min. The cytosolic lysate was loaded onto a non-linear sucrose gradient (from 5% to 34% to 55%) and ultracentrifuged at 220,000$g$ for 2 h at 4 °C, using the SW60 rotor in an Optima XPN-80 ultracentrifuge. Polysomal fractions, that is, efficiently translated mRNA (associated with >3 ribosomes), were collected using a piston gradient fractionator (BioComp), and RNA was extracted using the TriPure isolation method (11667157001, Roche) and kept at −80 °C. RNA was quantified using RiboGreen RNA Quantitation Kit (10207502, Thermo Fisher Scientific). The mRNA library preparation was performed using the TruSeq Stranded mRNA kit (20020595, Illumina). Sequencing was performed using the NovaSeq S1 (100 cycles) standard workflow.

## Assessment of protein synthesis

SENS or RES M395, MM074 and A375 cells were plated into 6-well plates in complete medium for 24 h. Newly translated proteins were assessed using the Click-iT Plus OPP Alexa Fluor 488 Protein Synthesis Assay kit (C10456, Thermo Fisher Scientific). Briefly, OP-puro was added at a concentration of 10 µM for 30 min. The cells were then trypsinized, washed and fixed with the Click-iT fixative. The pellet was then resuspended and permeabilized with a saponin-based permeabilization buffer and Click-iT Plus reaction cocktail containing a fluorescent picolyl azide dye. After washing and resuspension with the saponin-based permeabilization and wash reagent, the cells were analysed on the BD FACSCanto II System and quantified for the Alexa Fluor 488 fluorescence intensity. Three independent experiments were performed for each cell line.

For the HPG incorporation assay, M395 RES and A2058 cells depleted or not of VARS were incubated for 30 min at 37 °C in methionine-free media. HPG (C10186, Invitrogen) was added to the culture medium for 1 h at 50 µM of final concentration. The cells were then washed, fixed and permeabilized with 0.25% Triton X-100 in PBS. After washing, Alexa Fluor 488 azide (A10266, Invitrogen) was added using the Click-iT Cell Reaction buffer kit (C10269, Invitrogen), and the cells were analysed by flow cytometry.

## tRNA aminoacylation assay

The tRNA aminoacylation assay was used as previously described[32]. Briefly, SENS or RES cells, depleted or not for VARS, were lysed using the TriPure isolation method (11667157001, Roche) and precipitated overnight with 100% cold ethanol. The samples were then resuspended with acetate buffer (0.3 M, pH of 4.5) and 10 mM EDTA and precipitated with cold ethanol. A total of 2 µg of each RNA sample was then used to either add 10 mM of sodium periodate for the oxidized sample or 10 mM of sodium chloride for the control sample. After 20 min of incubation, the reactions were quenched with glucose. Yeast phenyl tRNA (R4018, Sigma-Aldrich) was added to each sample as an internal control. The samples were precipitated again with ethanol. The samples were then quenched with acetate buffer and precipitated. Finally, the samples were ligated to a 5′-adenylated DNA adaptor (5′-/5rApp/TGG AATTCTCGGGTGCCAAGG/3ddC/-3′), using truncated KQ mutant T4 RNA ligase 2 (M0373, New England Biolabs) overnight at 18°C.

Reverse transcription was performed with SuperScript IV Reverse Transcriptase (18090050, Thermo Fisher Scientific) according to the manufacturer's protocol, with a primer complementary to the DNA adaptor as in ref. 32. qPCR was then performed using tRNA-specific primers as described in ref. 32.

## esiRNA screen

Transfection of MISSION esiOPEN RNA was performed according to the manufacturer's protocol using DharmaFECT 2 Transfection Reagent in OptiMEM (Invitrogen). Fluorescence-activated cell sorting (FACS) analysis of PI staining was performed after 96 h of esiRNA transfection. A list of all esiOPEN gene studied is included in Supplementary Table 4.

## Western blot

Cells were lysed in lysis buffer (50 mM Tris–HCl pH 7.5, 1 mM EDTA, 1 mM EGTA, 1 mM sodium orthovanadate, 50 mM sodium fluoride, 0.27 M sucrose, 1% NP-40 and 1% 2-mercaptoethanol, with protease and phosphatase inhibitor). The protein quantification was measured using the BCA Protein Assay Kit (Thermo Fisher). For immunodetection, membranes were blocked in a blocking buffer solution and exposed to appropriate antibodies.

## Quantitative real-time PCR analysis

Total RNA was extracted using the EZNA Total RNA kit I (OMEGA Bio-Tek) according to the manufacturer's instructions. cDNA was synthesized using the RevertAid H Minus First Strand cDNA Synthesis kit (Thermo Fisher). Quantitative real-time PCR was performed in triplicates using TB Green premix Ex Taq (Takara) and a LightCycler 480 System (Roche).

## Immunohistochemistry and immunostaining assessment

The samples from patients with histologically confirmed primary ($n = 12$) and metastatic melanoma ($n = 21$) with or without adjacent normal tissue ($n = 12$) were retrieved from the biobank of the University Hospital Center in Liege. Informed consent was obtained from all patients providing samples and the protocol was approved by the ethical committee of the University of Liege (no. 3006695). A total of 4 µm sections of formalin-fixed paraffin-embedded specimens were first deparaffinized and rehydrated in graded alcohols, and the antigens were retrieved in citrate buffer (10 mM, pH 6) Endogenous peroxidases and non-specific binding sites were then blocked using 3% hydrogen peroxide and serum-free protein block reagent (Dako/Agilent Technologies), respectively. The sections were then incubated with primary antibody (VARS 1/200, low pH, NBP2-20843, Novus Biological; HADH 1/250, PA5-31157, Thermo Fisher Scientific) in Dako REAL Antibody Diluent buffer for 1 h at room temperature. After a washing step, the appropriate EnVision Detection system (Dako) was used in secondary reaction. The positive cells were finally visualized using alkaline phosphatase red chromogen. Whole slides were digitally scanned using a NanoZoomer 2.0 HT (Hamamatsu) and the immunolabelled melanoma cells were precisely quantified by computerized counts (QuPath 0.3.2)[54]. Briefly, after defining the optimal colour deconvolution, cell detection was conducted using the '« positive cell detection »' mode, and both the percentage of positive cells and staining intensity were determined using an arbitrary threshold of the red component of the staining. Low, moderate and strong intensities were considered for VARS expression. Regarding correlation analysis, the percentage of cells displaying strong VARS and HADH immunoreactivities were used using normal skin, primary and metastatic melanoma ($n = 22$ for VARS high and $n = 21$ for VARS low).

## Cell death measurement

Cells were plated in a 6-well plate and treated once for 24 h (for M395, MM029 and MM383) or 72 h (for A375 cells). After treatment, supernatant and trypsinized cells were collected and centrifuged for 5 min at 200$g$ at 4 °C. After washing with PBS and recentrifugation, the pellet was resuspended in a solution containing 50 µg ml⁻¹ PI (P4170, Sigma-Aldrich), 0.1% sodium citrate and 0.4% Triton X-100 for 2 h at 4 °C in the dark. Sub-G1 peak corresponding to the hypodiploid DNA content was quantified using the CytoFLEX Flow Cytometer (Beckman Coulter) as an indicator of cell death. Relative nuclear

fragmentation was calculated using the following formula: 1 + 100 times (the percentage of induced nuclear fragmentation – the percentage of spontaneous nuclear fragmentation) divided by (100 – the percentage of nuclear fragmentation). The data represent the mean + s.e.m. of three independent experiments carried out in duplicate.

### Establishment of VARS signature in patients with $BRAF^{V600E}$ mutation

HTSeq counts for the GDC TCGA SKCM cohort were downloaded from Xena Browser (https://xenabrowser.net). The patients were filtered on cBioPortal (SKCM (TCGA, PanCancer Atlas (https://www.cbioportal.org))) depending on their *BRAF* mutational status. The patients carrying at least one *BRAF* mutation were selected ($n = 233$) for further analysis. A score for VARS signature ($n = 84$) using the GSVA R package (v1.48.0)[55] was assessed using the ssGSEA method with a Poisson kernel. An analysis was performed comparing the patients with the 15% highest ($n = 35$) and 15% lowest ($n = 35$) score for VARS signature. Further cut-off was applied to remove genes that showed less than ten counts in more than 30% in both groups using edgeR's (v3.42.2)[56] filterByExpr method.

The data were then normalized to log (counts per million) using the Limma R package's (v3.56.1)[57] voom function with a design matrix based on the aforementioned group. The subsequent GSEA were performed using the GSEA software (v4.2.2)[58] and the gene sets from MsigDB.

### Mice experiments

Eight-week-old NOD/SCID mice (NOD.CB17-Prkdcscid/NCrCrl, *Mus musculus*, from Charles River) were subcutaneously inoculated with 1 million A375 RES cells depleted (shVARS#1 or shVARS#2) or not (shC-TRL) of VARS. After 4 days, the mice were treated by intraperitoneal injections with vemurafenib (25 mg kg$^{-1}$, once every 2 days). The mice were killed after 17 days, and tumour weight was assessed for all conditions. The weight of the mice was monitored throughout the whole experiment, and tumour size never exceeded 1 cm³.

The experiments were approved by the Ethical Committee of the University of Liege (no. 2126). The 'Guide for the Care and Use of Laboratory Animals,' prepared by the Institute of Laboratory Animal Resources, National Research Council and published by the National Academy Press, as well as European and local legislations, was followed carefully. Accordingly, the temperature and relative humidity were 21 °C and 45–60%, respectively. The cages were ventilated, softly lit and subjected to a light–dark cycle. The relative humidity was kept at 45–65%. The mouse rooms and cages were always kept at a temperature range of 20–24 °C.

### Lentiviral infections

Lenti-X 293T cells were transfected with 12 μg of the lentiviral construct of interest (shRNA or overexpressing plasmids), 5 μg of VSVG and 12 μg of pPsx2 using Mirus transfection reagent (Mirus TransIT-LT1 Transfection Reagent, MIR 2360, Fisher Scientific). All the constructs are listed in Supplementary Table 4. After 48 h, the supernatant was collected, centrifuged to remove cell debris and filtered through a 0.45 μm filter, and polybrene was added to the medium before transduction to target cells. The same procedure was repeated on the second day. The cells were then selected with puromycin (1 μg ml$^{-1}$) for 48 h. Knockdown or overexpression efficiency was confirmed by qPCR or western blot analysis. Three independent transductions were performed for all experiments.

### Ribosome immunoprecipitation

Ribosome immunoprecipitation with an RPL22-FLAG construct was performed according to the previously described method[59]. Briefly, the cells were first transduced with a control vector (LEGO) or RPL22-FLAG vector[31]. The cells were then depleted or not of VARS using an shRNA. After 15 min of treatment with CHX (100 μg ml$^{-1}$), cells were lysed in cell lysis buffer containing 20 mM HEPES KOH (pH 7.3), 150 mM KCl,

10 mM MgCl₂, 1% NP-40, EDTA-free protease inhibitors, 0.5 mM dithiothreitol, 100 μg ml$^{-1}$ of CHX, 10 μg ml$^{-1}$ RNasin and 10 μg ml$^{-1}$ of Superasin in RNase-free water, and the lysates were incubated overnight with anti-FLAG beads. After incubation, beads were washed extensively (five times with high-salt buffer and three times with low-salt buffer) and RNA was extracted using the EZNA Total RNA kit I (OMEGA Bio-Tek), according to the manufacturer's instructions.

### Ribosome profiling

Briefly, the cells depleted or not depleted of VARS were treated with CHX (0.1 mg ml$^{-1}$) for 1 min and then washed and lysed with a specific buffer. After RNase digestion, ribosome footprints were used to generate the RPF libraries using TuSeq Ribo Profile kits (Illumina) from three independent shRNA transductions. The sequencing of RPF libraries was performed using Illumina NextSeq500 sequencer to reach at least 30 million single-end reads for each sample. After adaptor cutting, quality trimming and removal of contaminants (rRNA and tRNA sequences), the cleaned reads were aligned to the human genome (GRCh38) using the nf-core RNA sequencing pipeline (v.3.0). Differential RPF expression analysis was performed with DESeq2. V.1.34.0 comparing shRNA VARS with shRNA CTRL (paired design). The diricore analysis was performed as in ref. 40.

### FA oxidation

FA oxidation activities were measured using a colorimetric FA oxidation assay kit (BR00001, AssayGenie). The assay was performed according to the manufacturer's protocol. The colorimetric reaction was red at 492 nm using a microplate reader.

### Statistics and reproducibility

Graphs were generated using Prism 10 (GraphPad) or R studio (2023.09.1 + 494). No statistical method was used to predetermine sample sizes. The data distribution was tested for normality before using parametric tests. After performing ROUT test on mice experiment, one mouse harbouring shCTRL tumour on one flank and M395 RES VARS-depleted (shVARS-1) tumour on the other flank was identified as an outlier and excluded from the analysis. The experiments were not randomized. The investigators were not blinded to allocation during experiments and outcome assessment.

All statistical tests were performed using Excel 2016 or Prism 10. The statistical analyses performed for each experiment are indicated in figure legends.

### Reporting summary

Further information on research design is available in the Nature Portfolio Reporting Summary linked to this article.

## Data availability

tRNA sequencing, RNA sequencing and polysome and ribosome sequencing data that support the findings of this study have been deposited in the Gene Expression Omnibus under accession codes GSE236645, GSE236046 and GSE236642, respectively.
MS data have been deposited in ProteomeXchange with the primary accession code PXD044863 and PXD044910.
The human melanoma data were derived from the TCGA Research Network can be found at http://cancergenome.nih.gov/. The dataset derived from this resource that supports the findings of this study is available in Xena Browser and on cBioPortal where patients were filtered depending on their BRAF mutational status.
All other data supporting the findings of this study are available from the corresponding author on reasonable request. Source data are provided with this paper.

## Code availability

All codes used in this study are available upon request.

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

## Acknowledgements

We thank G. Ghanem and A. Najem for the access to MM lines. We are grateful to the GIGA-imaging, bioinformatics, genomics and viral vector and the CHU Liege Biobank facilities. This study was supported by the Belgian National Funds for Scientific Research, Belgian Foundation against Cancer (2020-068; 2021-1593), Worldwide Cancer Research Association (23-0288) and Foundation Leon Fredericq. I.T. was supported by the Interdisciplinary Thematic Institute Integrative Molecular and Cellular Biology, part of the Interdisciplinary Thematic Institute 2021–2028 programme of the University of Strasbourg, CNRS and Inserm, (IdEx Unistra; ANR-10-IDEX-0002). P.C. is a Wel-RI Welbio investigator.

## Author contributions

Conceptualization and methodology: N.E.-H. and P.C. Software and formal analysis: N.E.-H., M.L., C.C., F.R., P.-R.K. and A.L. Investigation: N.E.-H., M.S.R., R.V, P.-R.K., K.S., L.M.-M., P.R, A.B., S.T., M.H. and E.G. Resources: A.C., B.P., L.N., I.T., P.T., J.C.-M., F.R., R.A. and P.C. Writing: N.E.-H. and P.C. Supervision and funding acquisition: P.C. All authors discussed the results and commented the paper.

## Competing interests

The authors declare no competing interests.

## Additional information

**Extended data** is available for this paper at https://doi.org/10.1038/s41556-024-01439-2.

**Correspondence and requests for materials** should be addressed to Pierre Close.

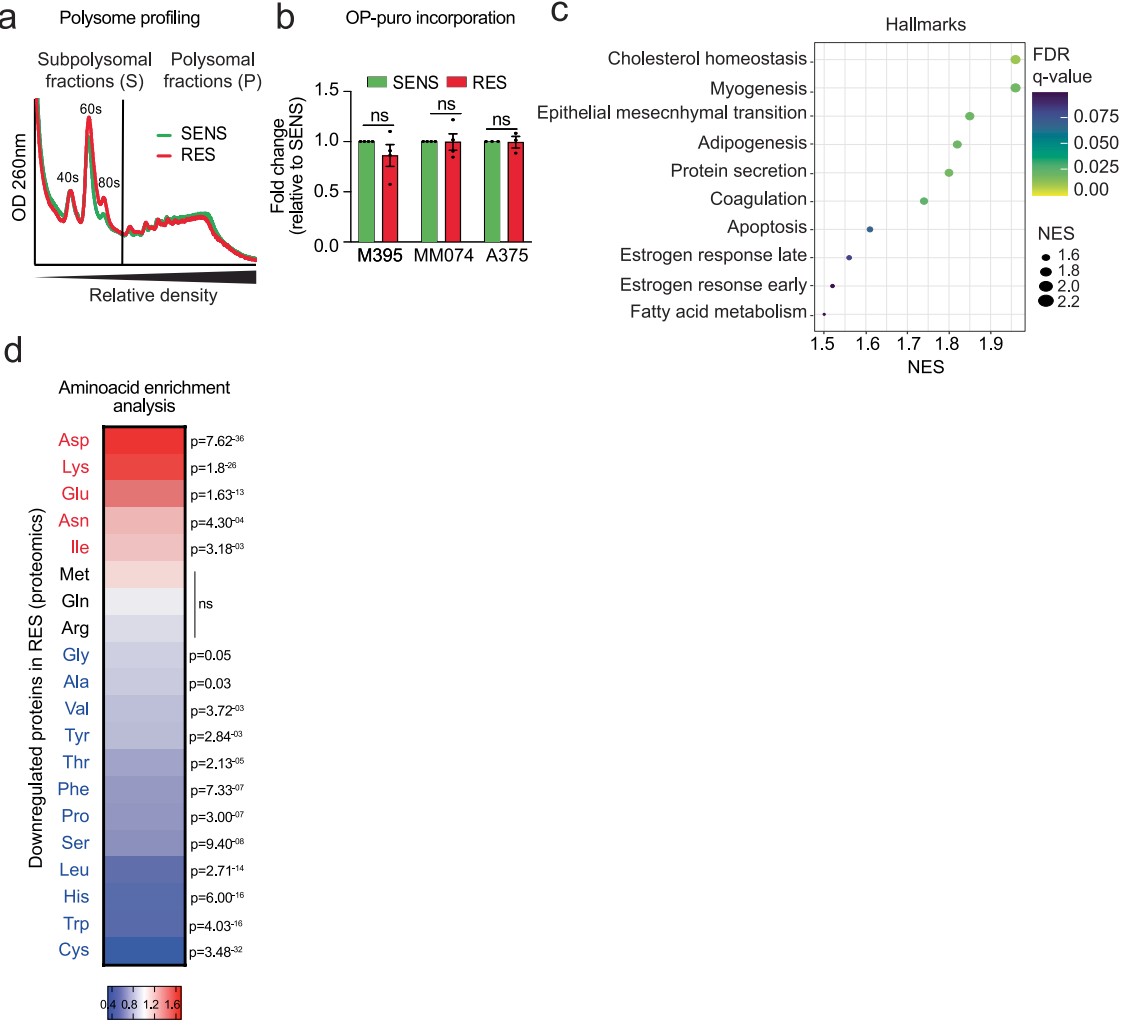

**Extended Data Fig. 1 | Overall translation rate is comparable in SENS and RES cultures. a**, Polysome profiling experiments of M395 SENS and RES cultures and **b**, Global rates of mRNA translation reported by total cell OP-puro fluorescence in M395, MM074 and A375 SENS and RES cultures (fold mean ± SEM of at least 3 independent replicates, two-tailed unpaired t-test was performed). **c**, Dot plot of significantly enriched terms in RES using the hallmark database (GSEA). The color of the bubbles represents the FDR q-value and the size of the bubbles represents the enrichment score (NES). **d**, Heatmap representing amino acids enrichment analysis of proteins downregulated in M395 RES compared to the global genome Significant differences are represented whereas red indicates an enrichment and blue an impoverishment (chi-squared test was performed). Colour bar, fold change.

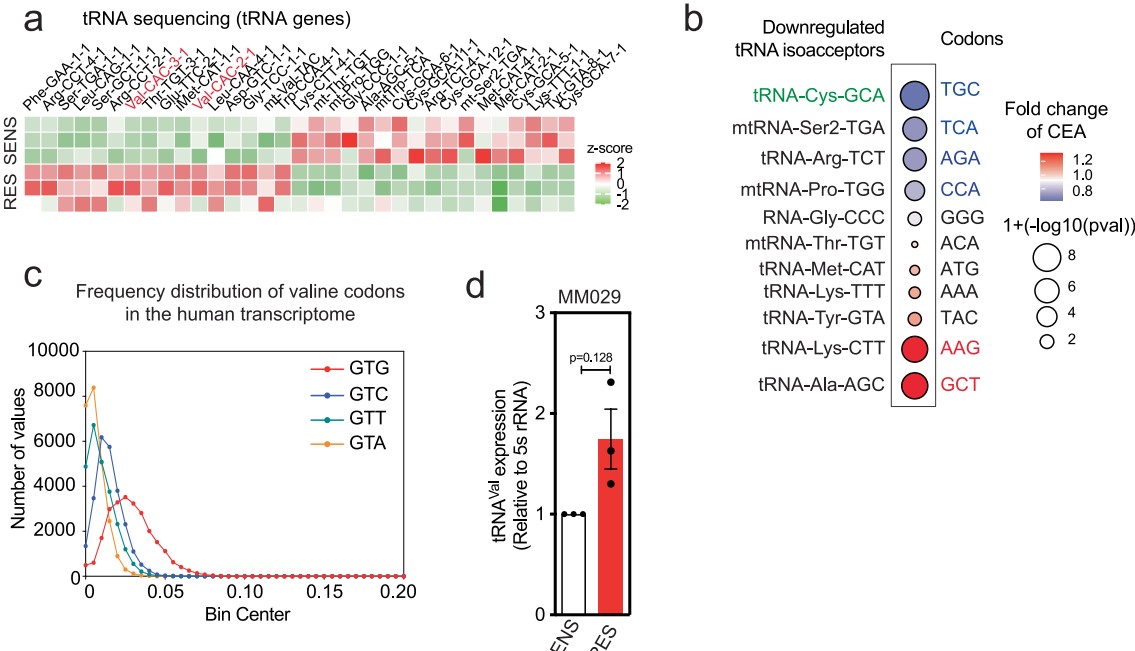

**Extended Data Fig. 2 | GTG codon is the most represented valine codon in the human transcriptome. a**, Heatmap representing the z-scores of significantly modified tRNA isodecoders (by tRNA sequencing) in M395 SENS and RES cultures (n = 3 independent replicates). tRNA expression profiling is annotated with gene symbol and anticodon. Red is used to highlight tRNA-Val-CAC isodecoder genes. **b**, Correlation between significant changes in codons from the CCA (chi-squared test) and the downregulated tRNA isoacceptors (Two-tailed unpaired t-test, no adjustment was used for these comparisons), Blue, impoverished codons and decrease in the corresponding isoacceptors tRNAs; red, enriched codons and the corresponding isoacceptors tRNAs; **c**, Histogram representing frequency distribution of valine codons in the human genome. **d**, Quantification of tRNA-Val-CAC in MM029 SENS and RES cultures (fold mean ± SEM of 3 independent replicates, two-tailed unpaired t-test was performed).

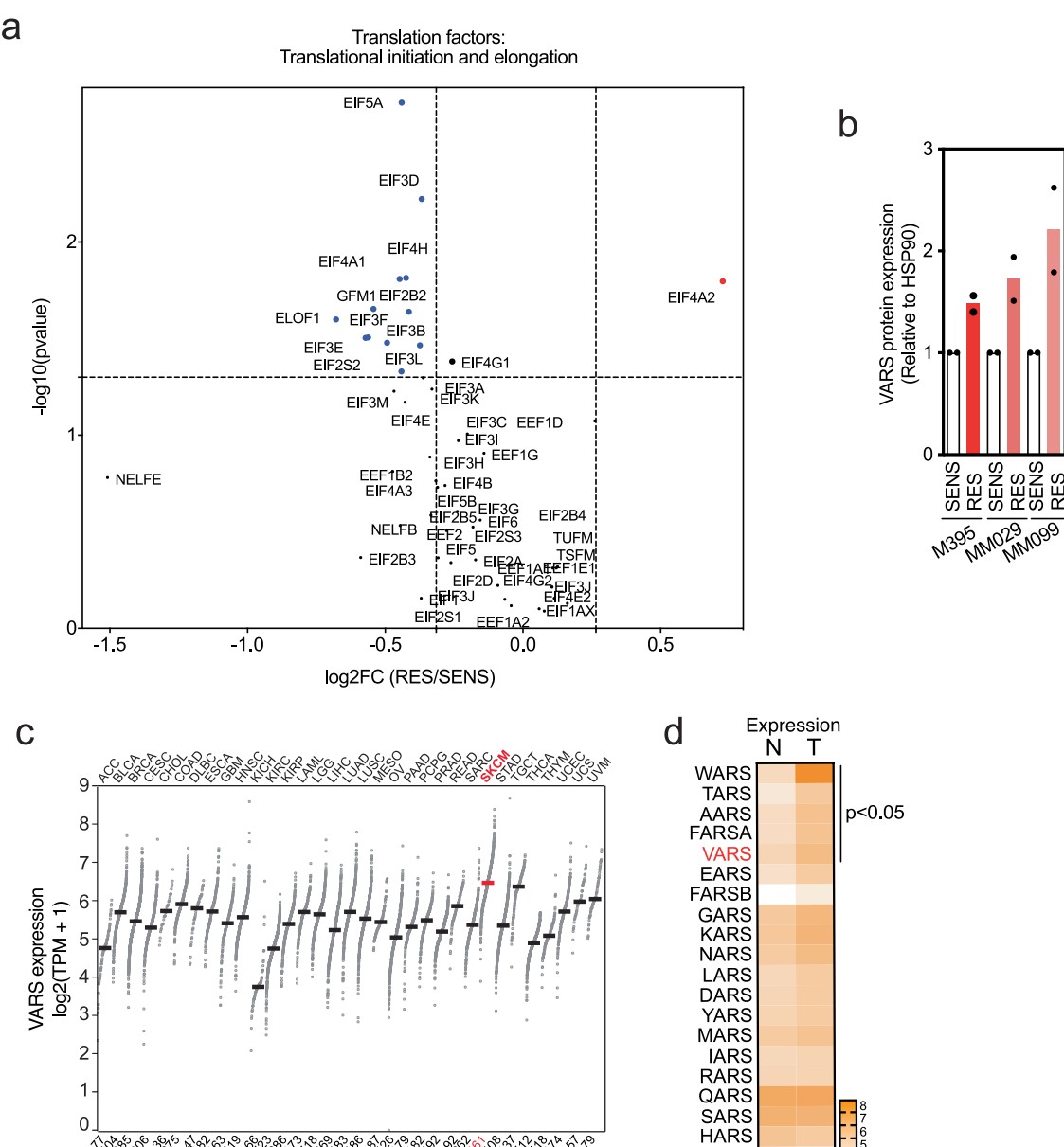

**Extended Data Fig. 3 | VARS expression correlates negatively with the survival of patients with melanoma. a**, Volcano plot representing translation factors (initiation and elongation) from M395 SENS and RES proteomics data. Significant changes (two-tailed unpaired t-test) are represented in red (upregulated genes) or in blue (downregulated genes). **b**, Quantification of VARS protein expression in M395, MM099 and MM029 SENS and RES cultures (Fold mean ± SEM of 2 independent replicates). **c**, VARS mRNA expression across cancers. TPM stands for transcripts per million. SKCM stands for skin cutaneous melanoma. **d**, TCGA analysis of all aminoacyl-tRNA synthetases expression in normal (N) and tumor (T) skin cutaneous melanoma. Pvalues were obtained from GEPIA.

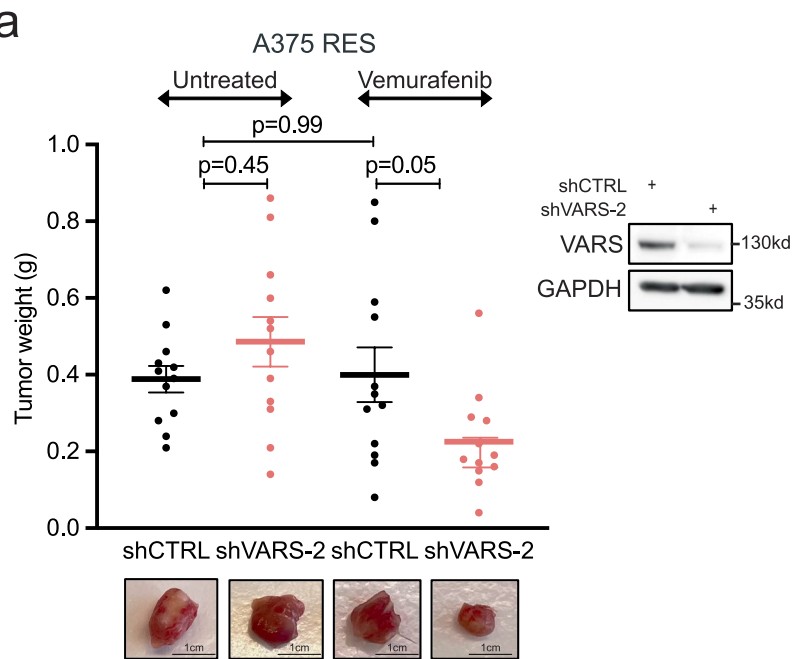

**Extended Data Fig. 4 | VARS depletion resensitizes melanoma tumors to MAPK-therapy. a**, A375 RES control or VARS-depleted melanoma cells (shVARS-2) were xenografted in mice and treated or not with Vemurafenib (25 mg/kg). Tumor weight was assessed and plotted (n = 12 for each group). Mean ± SEM is represented. Two-way ANOVA was performed. Tukey's multiple comparison tests are indicated in the figures.

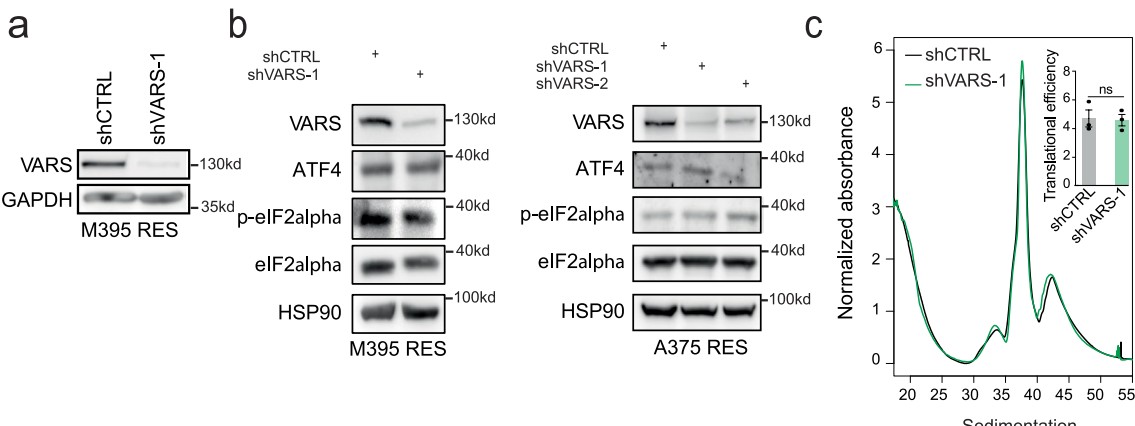

**Extended Data Fig. 5 | VARS depletion does not induce an integrated stress response response or demonstrate a global defect in translation. a**, Western blot corresponding to Fig. 5a, b. **b**, Western blot analyses of integrated stress response after VARS depletion. **c** Polysome profiles and translation efficiency of M395 RES control or depleted of VARS (fold mean ± SEM of n = 3 independent replicates, two-tailed unpaired t-test was performed).

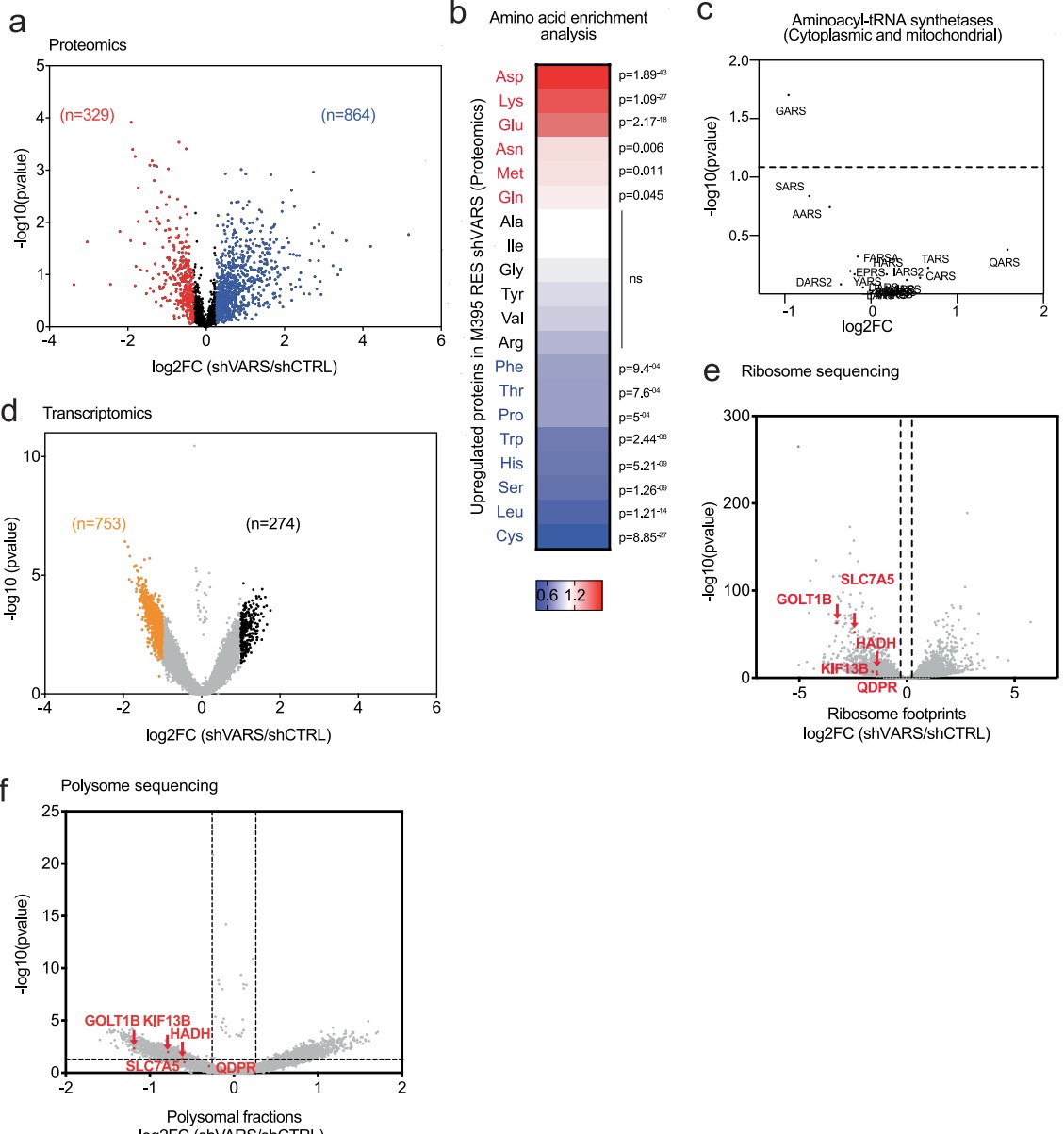

**Extended Data Fig. 6 | Proteomics analysis of VARS depleted cells reveals an enrichment in valine amino acid. a**, Volcano plots showing p-values (−log10) versus log2 fold change of proteomics analysis of melanoma M395 RES depleted for VARS as compared to the control (two-tailed unpaired t-test of n = 3 independent replicates), Blue: up-regulated; red: down-regulated proteins. **b**, Heatmap representing amino acids analysis of proteins upregulated in M395 RES depleted for VARS as compared to control. Chi-squared test was performed. Colour bar, fold change. **c**, Volcano plot listing detected aminoacyl-tRNA synthetases in proteomics analysis of M395 RES cells upon VARS depletion as compared to control (two-tailed unpaired t-test of n = 3 independent replicates). **d**, Volcano plots showing p-values (−log10) versus log2 fold change of transcriptomic analysis of melanoma M395 RES depleted for VARS as compared to the control; Black: up-regulated; Orange: down-regulated proteins (two-tailed unpaired t-test of n = 3 independent replicates). **e, f**, Volcano plots of ribosome (e) and polysome (f) sequencing highlighting the five identified candidates (HADH, KIF13B, SLC7A5, QDPR and GOLT1B) (two-tailed unpaired t-test of n = 3 independent replicates).

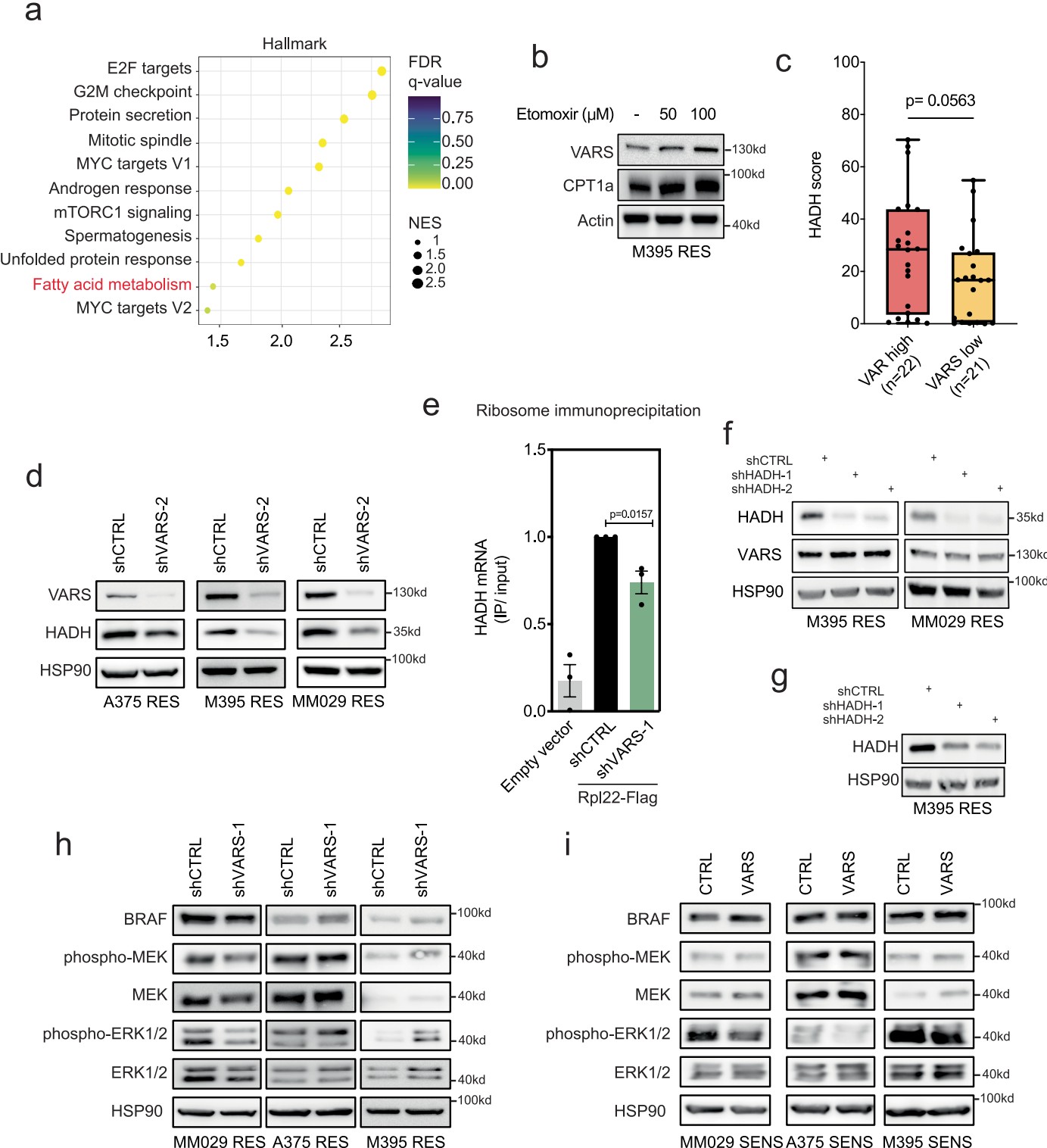

**Extended Data Fig. 7 | VARS correlates with HADH expression and VARS transcriptionally regulates HADH. a**, Dot plot of significantly enriched terms in VARS high versus VARS low TCGA melanomas using the hallmark database (GSEA). The color of the bubbles represents the FDR q-values and the size of the bubbles represents the enrichment scores (NES). **b**, Western blot analysis of M395 RES upon etomoxir treatment as indicated. CPT1a is used as a control (n = 3). **c**, HADH immunostaining score was represented for VARS high (n = 22) and VARS low patient (n = 21) groups in normal skin, primary and metastatic melanoma patient biopsies. Two-sided Mann-Whitney test was performed.

Whiskers are represented as min to max values with a median line. **d**, Western blot of VARS and HADH protein expression in A375, M395, MM029 RES cells depleted or not of VARS (shVARS-2). **e**, qRT-PCR after ribosomal immunoprecipitation using RPL22-Flag expression cells depleted (shVARS) or not (shCTRL) of VARS. Fold mean ± SEM of n = 3 independent replicates, two-tailed unpaired t-test was performed. See Fig. 7i. **f**, Western blot corresponding to Fig. 7i. **g**, Western blot corresponding to Fig. 7j (n = 3). **h,i**, Western blot showing MAPK pathway upon VARS depletion (h) or VARS overexpression (i) in MM029, A375 and M395 RES or SENS melanoma cells (n = 1).

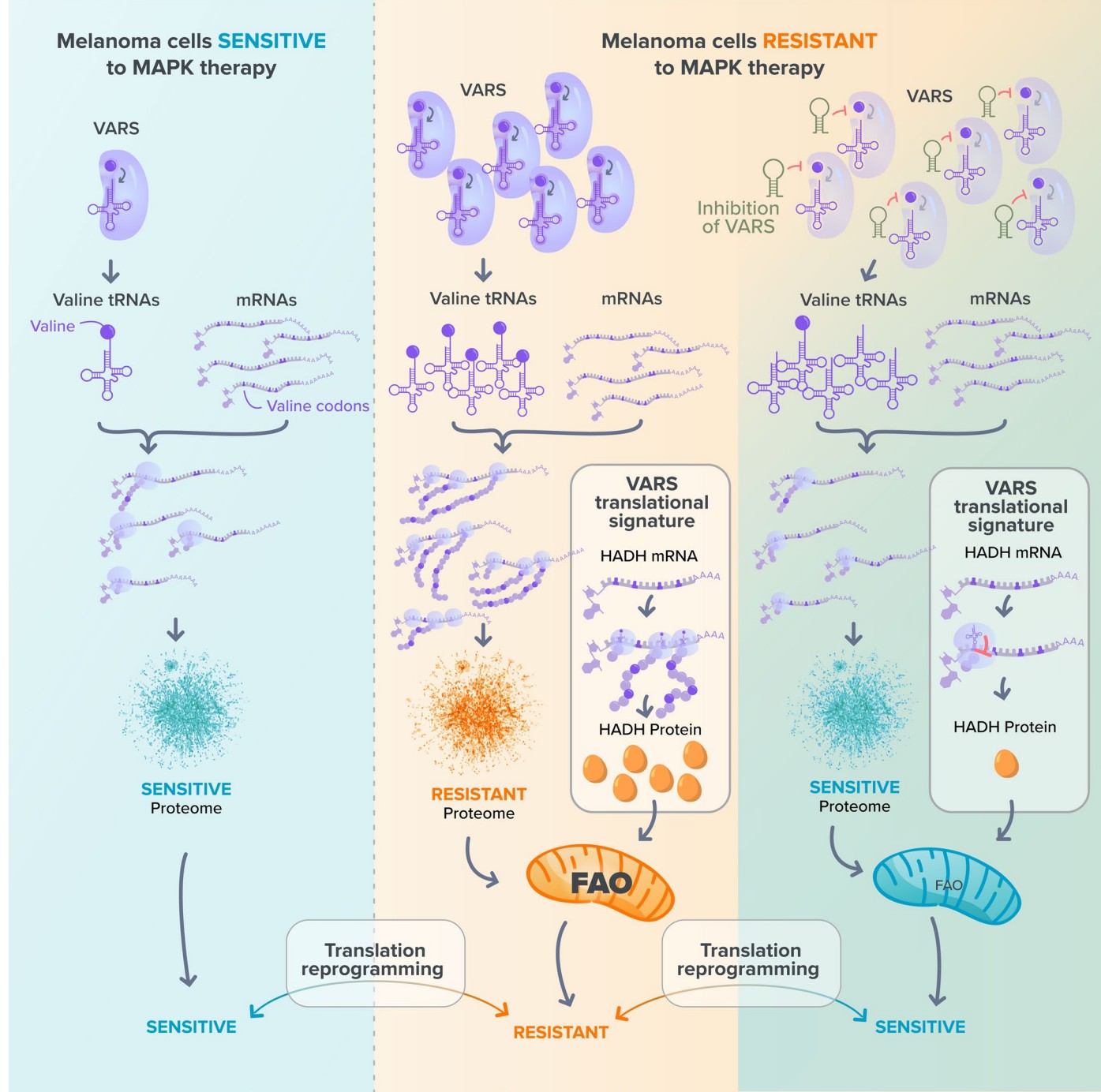

**Extended Data Fig. 8 | Model: VARS promotes melanoma resistance through translation of valine enriched genes, among them the fatty acid regulator HADH.**

# Reporting Summary

## Statistics

For all statistical analyses, confirm that the following items are present in the figure legend, table legend, main text, or Methods section.

| n/a | Confirmed | |
|---|---|---|
| ☐ | ☒ | The exact sample size (*n*) for each experimental group/condition, given as a discrete number and unit of measurement |
| ☐ | ☒ | A statement on whether measurements were taken from distinct samples or whether the same sample was measured repeatedly |
| ☐ | ☒ | The statistical test(s) used AND whether they are one- or two-sided<br>*Only common tests should be described solely by name; describe more complex techniques in the Methods section.* |
| ☐ | ☒ | A description of all covariates tested |
| ☐ | ☒ | A description of any assumptions or corrections, such as tests of normality and adjustment for multiple comparisons |
| ☐ | ☒ | A full description of the statistical parameters including central tendency (e.g. means) or other basic estimates (e.g. regression coefficient) AND variation (e.g. standard deviation) or associated estimates of uncertainty (e.g. confidence intervals) |
| ☐ | ☒ | For null hypothesis testing, the test statistic (e.g. *F*, *t*, *r*) with confidence intervals, effect sizes, degrees of freedom and *P* value noted<br>*Give P values as exact values whenever suitable.* |
| ☒ | ☐ | For Bayesian analysis, information on the choice of priors and Markov chain Monte Carlo settings |
| ☒ | ☐ | For hierarchical and complex designs, identification of the appropriate level for tests and full reporting of outcomes |
| ☒ | ☐ | Estimates of effect sizes (e.g. Cohen's *d*, Pearson's *r*), indicating how they were calculated |

*Our web collection on statistics for biologists contains articles on many of the points above.*

## Software and code

Policy information about availability of computer code

| | |
|---|---|
| Data collection | - Homo sapiens transcriptome was downloaded from the human genome database (HGD-ENSEMBL) for codon and amino acid enrichment analysis. |
| Data analysis | - All statistical tests: chi-square test, student's t test, ANOVA test were performed on Excel 2016 or GraphPad Prism 10.<br>- Codon enrichment was evaluated using a chi-squared test, comparing the codon distribution of each of the genes from each group to the human transcriptome. The same was applied for the amino acid enrichment analysis.<br>- Gene enrichment pathways were assessed by using Gene Set Enrichment Analysis (GSEA). Bubble plots representing the pathways were generated using R or GraphPad Prism.<br>- GEPIA2 tool was used to assess survival analysis and expression of aaRSs (in The Cancer Genome Atlas (TCGA) database).<br>- For the establishment of VARS signature: A custom R script using single-sample GSEA (ssGSEA) and a differential expression analysis with Limma R package's (v3.56.1) was used in this study.<br>- For proteomics analysis the following software were used: MaxQuant software v1.6.0.16<br>- QuPath 0.3.2 is used for the quantification of the immunolabelled melanoma cells.<br>- Diricore analysis for codon occupancy was used from github (https://github.com/pkorner218/Ribosome_Diricore_pipeline/). |

For manuscripts utilizing custom algorithms or software that are central to the research but not yet described in published literature, software must be made available to editors and reviewers. We strongly encourage code deposition in a community repository (e.g. GitHub). See the Nature Portfolio guidelines for submitting code & software for further information.

## Data

Policy information about availability of data

All manuscripts must include a data availability statement. This statement should provide the following information, where applicable:

- Accession codes, unique identifiers, or web links for publicly available datasets
- A description of any restrictions on data availability
- For clinical datasets or third party data, please ensure that the statement adheres to our policy

Polysome and RNA sequencing: Gene Expression Omnibus; accession number GSE236046;
Ribosome sequencing: Gene Expression Omnibus; accession number GSE236642;
tRNA sequencing: Gene Expression Omnibus; accession number GSE236645;

Proteomics data: The Mmass spectrometry data have been deposited in ProteomeXchange with the primary accession code PXD044863 (https://www.ebi.ac.uk/pride/archive/projects/PXD044863/private) and PXD044910 (https://www.ebi.ac.uk/pride/archive/projects/PXD044910/private).

For the establishment of VARS signature: A custom R script using single-sample GSEA (ssGSEA) and a differential expression analysis with Limma R package's (v3.56.1) was used in this study.

## Research involving human participants, their data, or biological material

Policy information about studies with human participants or human data. See also policy information about sex, gender (identity/presentation), and sexual orientation and race, ethnicity and racism.

| | |
|---|---|
| Reporting on sex and gender | 8 biopsies from women and 7 biopsies from men were available. Sex and gender were not considered in the study design nor in the data analysis. |
| Reporting on race, ethnicity, or other socially relevant groupings | N/A |
| Population characteristics | Human biopsy samples from male and female patients of age range between 26 and 81 years old were retrieved from the biobank of the University Hospital Center in Liege. Patients were all diagnosed with melanoma, and were tested positive for BRAF mutation. All samples used were obtained from leftover biopsy samples when available and did not interfere with standard practices of care (12 samples were obtained for normal skin, 12 samples for primary melanoma and 21 samples for metastatic melanoma). |
| Recruitment | No active recruitment was performed. Leftover biopsy samples were used when available. Informed consent was obtained from the patients providing samples. The participants were not compensated. |
| Ethics oversight | The protocol was approved by the ethical committee of the University of Liege (#3006695). |

Note that full information on the approval of the study protocol must also be provided in the manuscript.

# Field-specific reporting

Please select the one below that is the best fit for your research. If you are not sure, read the appropriate sections before making your selection.

☒ Life sciences          ☐ Behavioural & social sciences          ☐ Ecological, evolutionary & environmental sciences

For a reference copy of the document with all sections, see nature.com/documents/nr-reporting-summary-flat.pdf

# Life sciences study design

All studies must disclose on these points even when the disclosure is negative.

| | |
|---|---|
| Sample size | Animal sample size experiment was assessed by Web Power. For the other experiments, no sample size calculation was performed. |
| Data exclusions | Concerning mice experiment, we performed ROUT test on GraphPad Prism to identify outliers. One mouse harbouring shCTRL tumor on one flank and shVARS-1 tumor on the other flank was identified as outlier and excluded from the analysis. |
| Replication | All experiments were performed with at least 2 biological replicates. The exact number of replicates is stated in figure legends |
| Randomization | Randomization was not performed, experiments were done in cell lines or xenografts were randomization is not applicable. Allocation of samples into experimental groups is not relevant, as the study helps us identify new features to discrimnate between already established experimental groups. |
| Blinding | Xenograft experiments were not performed in blind as these experiments are conducted for explorative purposes. |

# Reporting for specific materials, systems and methods

We require information from authors about some types of materials, experimental systems and methods used in many studies. Here, indicate whether each material, system or method listed is relevant to your study. If you are not sure if a list item applies to your research, read the appropriate section before selecting a response.

## Materials & experimental systems

| n/a | Involved in the study |
|---|---|
| ☐ | ☒ Antibodies |
| ☐ | ☒ Eukaryotic cell lines |
| ☒ | ☐ Palaeontology and archaeology |
| ☐ | ☒ Animals and other organisms |
| ☒ | ☐ Clinical data |
| ☒ | ☐ Dual use research of concern |
| ☒ | ☐ Plants |

## Methods

| n/a | Involved in the study |
|---|---|
| ☒ | ☐ ChIP-seq |
| ☐ | ☒ Flow cytometry |
| ☒ | ☐ MRI-based neuroimaging |

## Antibodies

| | |
|---|---|
| Antibodies used | All antibodies and dilutions used in this study are listed in supplementary table 1.<br>Name Antibody Host Reactivity Company Catalogue number Size (kDa) Application<br>VARS ValRS (D-7) Mouse H, M, Rats Santa-Cruz sc-166674 130 WB<br>NBP2-20843 VARS Rabbit H Novus Biological NBP2-20843  IHC<br>HADH HADH Rabbit H, M, Rats Invitrogen/ Thermofisher PA5-31157 34 WB/ IHC<br>CPT1a CPT1A (D3B3) Rabbit H Cell Signaling 12252 88 WB<br>GAPDH GAPDH (D16H11) Rabbit H, M, R, Mk Cell Signaling 5174 37 WB<br>HSP90 Hsp90 alpha/beta (H-114) Rabbit H Santa-Cruz sc-7947 90 WB<br>ATF4 ATF4  Rabbit H, M, R Cell Signaling 11815s 49 WB<br>p-EIF2alpha (Ser51) Phospho-eIF2α (Ser51) Rabbit H,M, R Cell Signaling 9721S 38 WB<br>EIF2alpha  eIF2α Rabbit H,M, R Cell Signaling 9722 38 WB<br>SLC7A5 LAT1 Rabbit H Cell Signaling 5347S 39 WB<br>KIF13B KIF13B Rabbit H Bio-techne NBP1-83398 200 WB<br>QDPR QDPR (B-1) Mouse H Santa-Cruz sc-376218 26 WB<br>GOLT1B GOLT1B Rabbit H, M, R Gentaur DF9071 15 WB<br>BRAF (F-3), BRAF, Mouse, H,M,Rats, Santa cruz, sc-55522, 95, WB<br>MEK1/2 (L38C12), MEK 1/2,  Mouse, H, M, R, Mk, Cell Signaling, 4694, 45, WB<br>Phospho-MEK1/2 (Ser217/221) (41G9), Phospho-MEK1/2, Rabbit, H, M, R, Mk, Cell Signaling, 9154, 45, WB<br>p44/p42 MAPK (Erk1/2),  ERK, Rabbit, H, M, R, Hm, Mk, Mi, Z, B, Pg, Sc, Cell Signaling, 9102, 42-44, WB/IP<br>Phospho-p44/42 MAPK (Erk1/2) (Thr202/Tyr204), Phospho-ERK, Rabbit, H, M, R, Hm, Mk, Mi, Z, B, Pg, Sc, , Cell Signaling, 4370, 42-44, WB/ IP/ IHC/ IF/ F |
| Validation | Validation of the listed antibodies was performed by the manufacturer. Data is available at the manufacturer's website as indicated below:<br>ValRS sc-166674: https://www.scbt.com/fr/p/valrs-antibody-d-7<br>VARS (NBP2-20843): https://www.novusbio.com/products/vars-antibody_nbp2-20843<br>HADH (PA5-31157): https://www.thermofisher.com/antibody/product/HADH-Antibody-Polyclonal/PA5-31157<br>CPT1a (12252): https://www.cellsignal.com/products/primary-antibodies/cpt1a-d3b3-rabbit-mab/12252<br>GAPDH (5174): https://www.cellsignal.com/products/primary-antibodies/gapdh-d16h11-xp-rabbit-mab/5174<br>HSP90 (sc-7947): https://www.scbt.com/p/hsp-90alpha-antibody-f-2?gad_source=1&gclid=EAIaIQobChMIopaDxem5hQMVnLVoCR1Z_QqREAAYASAAEgJgWPD_BwE<br>ATF4 (11815s): https://www.cellsignal.com/datasheet.jsp?productId=11815&images=1<br>p-EIF2alpha (Ser51) (9721S): https://www.cellsignal.com/products/primary-antibodies/phospho-eif2a-ser51-antibody/9721<br>EIF2alpha  (9722): https://www.cellsignal.com/products/primary-antibodies/eif2a-antibody/9722<br>SLC7A5 (5347S): https://www.cellsignal.com/products/primary-antibodies/lat1-antibody/5347<br>KIF13B (NBP1-83398): https://www.bio-techne.com/p/antibodies/kif13b-antibody_nbp1-83398<br>QDPR (sc-376218): https://www.scbt.com/fr/p/qdpr-antibody-b-1<br>BRAF (F-3) (sc-55522): https://www.scbt.com/p/raf-b-antibody-f-3?gad_source=1&gclid=EAIaIQobChMImeTrrOu5hQMVqS0GAB1hwgDzEAAYASAAEgLaj_D_BwE<br>MEK1/2 (L38C12) (4694): https://www.cellsignal.com/products/primary-antibodies/mek1-2-l38c12-mouse-mab/4694<br>Phospho-MEK1/2 (Ser217/221) (41G9) (9154): https://www.cellsignal.com/products/primary-antibodies/phospho-mek1-2-ser217-221-41g9-rabbit-mab/9154<br>p44/p42 MAPK (Erk1/2) (9102): https://www.cellsignal.com/products/primary-antibodies/p44-42-mapk-erk1-2-antibody/9102<br>Phospho-p44/42 MAPK (Erk1/2) (Thr202/Tyr204) (4370): https://www.cellsignal.com/products/primary-antibodies/phospho-p44-42-mapk-erk1-2-thr202-tyr204-d13-14-4e-xp-174-rabbit-mab/4370<br><br>For GOLT1B antibody validation was performed in Liu et al. Cancer Cell Int (2021). |

# Eukaryotic cell lines

Policy information about cell lines and Sex and Gender in Research

| | |
|---|---|
| Cell line source(s) | Cell lines source is stated in the material and methods section of the manuscript (Cell culture).<br>A375 melanoma cell lines were purchased from ATCC.<br>A375 vemurafenib-resistant cells were generated by increasing doses of vemurafenib up to 1 µM.<br>M395 (SENS and 1 µM vemurafenib RES) lines were from the laboratory of R. Lo (UCLA Division of Dermatology) - non commercial.<br>MM029, MM099 and MM383 naïve and their resistant counterparts (0.2 µM dabrafenib and 40 nM of trametinib) were provided by JC. Marine (KU Leuven) - non commercial<br>MM074 was provided by G. Ghanem (Institut J. Bordet, Université Libre de Bruxelles) - non commercial<br>Lenti-X293T cells were purchased from sigma-aldrich |
| Authentication | None of the cell lines used were authenticated. |
| Mycoplasma contamination | Mycoplasma test was performed routinely. Only negative lines are used in the study. |
| Commonly misidentified lines<br>(See ICLAC register) | None of the used cells are in the misidentified lines. |

# Animals and other research organisms

Policy information about studies involving animals; ARRIVE guidelines recommended for reporting animal research, and Sex and Gender in Research

| | |
|---|---|
| Laboratory animals | Eight-weeks-old NOD/SCID mice. Experiments were approved by the Ethical committee of the University of Liege (#2126). The temperature and relative humidity were 21°C and 45–60%, respectively. Cages were ventilated, softly lit, and subjected to a light dark cycle. |
| Wild animals | no wild animal was used in thids study |
| Reporting on sex | Sex was not considered in the study design. |
| Field-collected samples | no field collected samples were used in the study. |
| Ethics oversight | Experiments were approved by the Ethical committee of the University of Liege (#2126). |

Note that full information on the approval of the study protocol must also be provided in the manuscript.

# Flow Cytometry

## Plots

Confirm that:

☒ The axis labels state the marker and fluorochrome used (e.g. CD4-FITC).

☒ The axis scales are clearly visible. Include numbers along axes only for bottom left plot of group (a 'group' is an analysis of identical markers).

☒ All plots are contour plots with outliers or pseudocolor plots.

☒ A numerical value for number of cells or percentage (with statistics) is provided.

## Methodology

| | |
|---|---|
| Sample preparation | For the nuclear fragmentation assay: Supernatant and trypsinized cells were collected and stained with Nicoletti Buffer (0,1% Trinatriumcitrat-Dihydrat pH7.4, 0,1% Triton-X 100, 0.01% Propidium iodide).<br>For OPP and HPG assay: Cells were incubated with OPP or HPG (following an incubation in methionine-free media for HPG analysis) at indicated times in the material and methods section. Cells were then washed, fixed and permeabilized and the Click-iT Cell Reaction buffer Kit was used following the manufacturer's instructions. |
| Instrument | FACS Canto II (OPP and HPG), Cytoflex Cytometer (Nuclear fragmentation assay) |
| Software | FlowJo, BD FACSDiva software. |
| Cell population abundance | No sorting was performed |

Gating strategy

The gating strategy for cell death assay, OPP and HPG are provided in supplementary information.
Cell death was determined as sub G1 population.
For HPG and OPP healthy cells were gated (FSC_SSC plot) and doublets were removed (FSC-A_FSC-H plot). Cells positive for OPP or HPG are represented in the gating strategy.

☒ Tick this box to confirm that a figure exemplifying the gating strategy is provided in the Supplementary Information.

nature portfolio | reporting summary

April 2023

