## [Peer Review File · Nature Cell Biology]

Peer Review Information

Journal: Nature Cell Biology

Manuscript Title: Valine aminoacyl-tRNA synthetase promotes therapy resistance in melanoma

Corresponding author name(s): Pierre Close

Editorial Notes:

Reviewer Comments & Decisions:

Decision Letter, initial version:
--

*Please delete the link to your author homepage if you wish to forward this email to co-authors.

Dear Professor Close,

Your manuscript, "Valine aminoacyl-tRNA synthetase promotes therapy resistance in melanoma", has now been seen by 3 referees, who are experts in tRNA, cancer, translational control (referee 1); tRNA biology and function (referee 2); and melanoma, targeted therapy (referee 3). As you will see from their comments (attached below) they find this work of potential interest, but have raised substantial concerns, which in our view would need to be addressed with considerable revisions before we can consider publication in Nature Cell Biology.

Nature Cell Biology editors discuss the referee reports in detail within the editorial team, including the chief editor, to identify key referee points that should be addressed with priority, and requests that are overruled as being beyond the scope of the current study. To guide the scope of the revisions, I have listed these points below. We are committed to providing a fair and constructive peer-review process, so please feel free to contact me if you would like to discuss any of the referee comments further.

In particular, it would be essential to:

A) Strengthen the proposed mechanism as suggested by Reviewer 3:

"While the study focused on broader effects of VARS on mRNA translation, it does not address if VARS alterations in the BRAFi-resistant melanomas could influence the mutant BRAF activated MAPK pathway itself, which has been shown in multiple studies, to be altered in BRAFi-resistant melanomas resulting in resistance. A simple western blotting showing the effect of VARS knockdown or upregulation on the protein levels of BRAF, MEK, Phospho-MEK, MAPK (ERK1/2) and Phospho-MAPK (ERK1/2) would help understand its potential specific effects on the MAPK pathway itself."

"While I know that A375 and M395 cells are homozygous for BRAF-V600E, I do not know about the other cell lines in the study. This may be true for most readers. It is important to understand if there is heterozygosity in any of the cell lines in this study, and if BRAFi/MEKi treatment could potentially be selecting for BRAF-WT transcripts in the resistant cells, which could be happening at level of transcription or translation. This is especially important, as the cell lines are not consistently used in the experiments. Different cell lines were used for different experiments in the study."

"On the same lines as point 2 above, I am intrigued to know if VARS upregulation may result in selective complete translation BRAF-WT transcripts at a higher rate than the mutant-BRAF transcripts. This could be important to know because after all, it is the Valine amino acid that is substituted out in BRAF-V600E mutants. Also, what is the possibility that VARS upregulation may result in mis-incorporation of Valine instead of Glutamic acid in the 600 aa position of BRAF?"

B) Improve data rigor and interpretation as questioned by all reviewers:

Reviewer 1

"The key finding of Valine tRNA modulation need to be validated by Northern, as this is central to the story and Northern analysis is the gold standard."

"The in vivo finding in Fig. 4f of VARS depletion synergizing with Vemurafinib needs to be repeated with an independent shRNA to ensure lack of off-target effects, as this is a key and important discovery in the manuscript. The western blots showing HADH protein abundance reduction also needs to be repeated with an additional VARS shRNA."

"The critical conclusion is that VARS upregulation and enhanced valine tRNA promote translation of HADH in a codon-dependent manner needs to be demonstrated using a codon-specific reporter. This is critical and will demonstrate codon-dependent effects and assuage future nay-sayers who will claim that a direct mutagenesis experiment was not done to exclude non-direct effects on the HADH target gene."

Reviewer 2

"As a study dealing with tRNA synthetase and tRNA expression and charging, the biochemical characterization does not reflect the state-of-the-art of the field. The authors used something like

hydro-seq from 2017 which relies on sequencing fragmented tRNAs. The authors seem to have performed demethylase treatment to remove m1A and m3C, but the efficiency of this step was not described. In any case, multiple new tRNA-seq procedures have been published since 2020 which can provide more precise information on tRNA expression.

Fig. 2: The inadequate sequencing likely reflects the fact that qRT-PCR shows >4-fold change, but sequencing shows maybe 1.5-fold change in tRNA^{Val} expression. The authors should validate tRNA^{Val} expression results by Northern blot, not by qRT-PCR because of the differences in RT-interfering modifications."

"Fig. 3, 4: tRNA charging was measured using a complicated RT-PCR method without adequate controls. The standard method to directly and unambiguously measure tRNA charging is by acid-denaturing gel electrophoresis. There are now non-radioactive methods for tRNA Northern blots using biotinylated probes for charging measurements, no radioactivity needed."

"Fig. 5: how do cells still do well in translation after severe knockdown of an essential house-keeping gene like VARS? I am surprised to see that the VARS depletion does not trigger ISR. I am also surprised to see that ribosome does not seem to stall at Val codons in the A site. Does this mean the polysome is still fully loaded with charged tRNA^{Val} despite the decrease of charged tRNA^{Val} levels in the total tRNA?"

Reviewer 3

"While I know that A375 and M395 cells are homozygous for BRAF-V600E, I do not know about the other cell lines in the study. This may be true for most readers. It is important to understand if there is heterozygosity in any of the cell lines in this study, and if BRAFi/MEKi treatment could potentially be selecting for BRAF-WT transcripts in the resistant cells, which could be happening at level of transcription or translation. This is especially important, as the cell lines are not consistently used in the experiments. Different cell lines were used for different experiments in the study."

"The concentrations of BRAF and MEK inhibitors used in this study are too high. Vemurafenib at doses of 10 and 20 micro molar, and Trametinib at a dose of 1 micro molar could have non-specific effects. Also, the treatment times are not indicated in the nuclear fragmentation assays. The methods section suggests 24h, 48h and 72h. What times do the data represent?"

Additionally, different cell lines show sensitivity to BRAFi and MEKi at different times. For example, A375 may show significant nuclear fragmentation after 48h treatment, while others may show at 24h or 72h. So the combination of single time of treatment and extremely high doses of drugs makes the nuclear fragmentation data somewhat un-interpretable.

MTT based or Cell proliferation (growth) data showing dose-dependent effects on the cell lines used in the study could improve interpretation of the molecular data in the different cell lines. This could also help to address the concern about different cell lines for different experiments in point 2."

C) All other referee concerns pertaining to strengthening existing data, providing controls, methodological details, clarifications and textual changes as applicable should also be addressed.

D) Finally please pay close attention to our guidelines on statistical and methodological reporting (listed below) as failure to do so may delay the reconsideration of the revised manuscript. In particular please provide:

We would be happy to consider a revised manuscript that would satisfactorily address these points, unless a similar paper is published elsewhere, or is accepted for publication in Nature Cell Biology in the meantime.

- ensure that it conforms to our format instructions and publication policies (see below and www.nature.com/nature/authors/).

- provide a point-by-point rebuttal to the full referee reports verbatim, as provided at the end of this letter.

- provide the completed Editorial Policy Checklist (found here <https://www.nature.com/authors/policies/Policy.pdf>), and Reporting Summary (found here <https://www.nature.com/authors/policies/ReportingSummary.pdf>). This is essential for reconsideration of the manuscript and these documents will be available to editors and referees in the event of peer review. For more information see <http://www.nature.com/authors/policies/availability.html> or contact me.

Nature Cell Biology is committed to improving transparency in authorship. As part of our efforts in this direction, we are now requesting that all authors identified as 'corresponding author' on published papers create and link their Open Researcher and Contributor Identifier (ORCID) with their account on

the Manuscript Tracking System (MTS), prior to acceptance. ORCID helps the scientific community achieve unambiguous attribution of all scholarly contributions. You can create and link your ORCID from the home page of the MTS by clicking on 'Modify my Springer Nature account'. For more information please visit www.springernature.com/orcid.

[REDACTED]

We would like to receive a revised submission within six months. We would be happy to consider a revision even after this timeframe, however if the resubmission deadline is missed and the paper is eventually published, the submission date will be the date when the revised manuscript was received.

We hope that you will find our referees' comments, and editorial guidance helpful. Please do not hesitate to contact me if there is anything you would like to discuss.

Best wishes,

Zhe Wang

Zhe Wang, PhD
Senior Editor
Nature Cell Biology

Tel: +44 (0) 207 843 4924
email: zhe.wang@nature.com

Reviewers' Comments:

Reviewer #1:
Remarks to the Author:

In this manuscript, El-Hachem and colleagues find that melanoma cells that have acquired resistance to MAPK inhibition therapy upregulate expression of proteins enriched in valine amino acids. They further demonstrate that Valine tRNA and the VARS valine aminoacyl synthetase enzyme are upregulated in these cells. They go on to show that depletion of VARS sensitizes melanoma cells to MAPK inhibition therapy. They go on to identify the fatty acid oxidation gene HADH as a valine-enriched translationally upregulated protein that mediates resistance to MAPK inhibition. These findings reveal specific tRNA/charging enzyme modulation to underlie resistance to a widely clinically used therapeutic and extend in a meaningful way—with therapeutic significance and novelty—on previous studies that have demonstrated tRNA/charging enzyme modulation roles in cancer progression. The work is well done, novel, rigorous and addresses an important basic and clinically significant problem of therapeutic resistance. The manuscript is well-written, concise and clear. I have only a few, yet critical recommendations below that need to be addressed prior to publication in NCB.

1. The key finding of Valine tRNA modulation need to be validated by Northern, as this is central to the story and Northern analysis is the gold standard.
2. The in vivo finding in Fig. 4f of VARS depletion synergizing with Vemurafinib needs to be repeated with an independent shRNA to ensure lack of off-target effects, as this is a key and important discovery in the manuscript. The western blots showing HADH protein abundance reduction also needs to be repeated with an additional VARS shRNA.
3. The critical conclusion is that VARS upregulation and enhanced valine tRNA promote translation of HADH in a codon-dependent manner needs to be demonstrated using a codon-specific reporter. This is critical and will demonstrate codon-dependent effects and assuage future nay-sayers who will claim that a direct mutagenesis experiment was not done to exclude non-direct effects on the HADH target gene.

Reviewer #2:

Remarks to the Author:

This work describes the role of a specific aminoacyl-tRNA synthetase in the therapy resistance of melanoma. Aminoacyl-tRNA synthetases have been studied for a long time as house-keeping genes that are required for protein synthesis and cell growth. Recently, numerous studies have shown that specific tRNA synthetases play various roles in different types of cancer, either as promoters or suppressors. The cancer-specific roles of AARS are generally derived from their effects on cognate tRNA charging that are important to fine-tune translation of specific genes required for relevant cancer processes such as metastasis and others. This work shows that valine tRNA synthetase (VARS) plays a promotional role in the codon-dependent translational reprogramming induced by resistance to targeted MAPK therapy in melanoma. When melanoma becomes drug resistant, their proteomes are biased toward the usage of valine, which is also accompanied by valine-tRNA and VARS overexpression. The authors also identify specific pathways involved in the resistance, such as an enzyme in fatty acid oxidation. They also show

that resistant melanoma cultures are addicted to fatty acid oxidation and this enzyme for survival upon drug treatment. This work adds the growing literature on specific tRNA acting enzymes in specific cancers, but with the exiting new twist on finding the specific metabolic pathways required, as well as finding an essential house-keeping gene as cancer target in this specific context.

1. As a study dealing with tRNA synthetase and tRNA expression and charging, the biochemical characterization does not reflect the state-of-the-art of the field. The authors used something like hydro-seq from 2017 which relies on sequencing fragmented tRNAs. The authors seem to have performed demethylase treatment to remove m1A and m3C, but the efficiency of this step was not described. In any case, multiple new tRNA-seq procedures have been published since 2020 which can provide more precise information on tRNA expression.

Fig. 2: The inadequate sequencing likely reflects the fact that qRT-PCR shows >4-fold change, but sequencing shows maybe 1.5-fold change in tRNA^{Val} expression. The authors should validate tRNA^{Val} expression results by Northern blot, not by qRT-PCR because of the differences in RT-interfering modifications.

2. Fig. 3, 4: tRNA charging was measured using a complicated RT-PCR method without adequate controls. The standard method to directly and unambiguously measure tRNA charging is by acid-denaturing gel electrophoresis. There are now non-radioactive methods for tRNA Northern blots using biotinylated probes for charging measurements, no radioactivity needed.

3. Fig. 5: how do cells still do well in translation after severe knockdown of an essential house-keeping gene like VARS? I am surprised to see that the VARS depletion does not trigger ISR. I am also surprised to see that ribosome does not seem to stall at Val codons in the A site. Does this mean the polysome is still fully loaded with charged tRNA^{Val} despite the decrease of charged tRNA^{Val} levels in the total tRNA?

Reviewer #3:

Remarks to the Author:

This study by Najla El-Hachem et al is a well-written, interesting and extensive investigation of the role of Valine amino acyl tRNA synthetase (VARS) in BRAF inhibitor (BRAFi)-resistant melanomas. My comments are specifically focused on interpretation of the cellular and molecular effects of VARS alterations in the melanoma cells, tumors and clinical samples.

The discovery of the BRAF-V600E mutation as a driving force in melanoma oncogenesis and growth led to the development of BRAF-V600 mutant specific targeted inhibitors and MEK inhibitors targeting the BRAF-activated MAPK pathway. Subsequent clinical studies showed dramatic efficacy of these drugs at low nanomolar doses, but also showed that the efficacy is short-lived due to development of resistance. Many studies have since shown different mechanisms of resistance to these inhibitors. Some results

from this study parallel other studies which have shown the role of metabolic pathways in resistance, for example the role of Fatty acid oxidation and its key enzymes. However, the mechanisms of these broad pathway-dependent resistances are poorly defined. It is possible that VARS may play an important role in the generation of such resistance. While this study is compelling in that context, there are several questions and experimental concerns as described below. Addressing these concerns could potentially improve the study.

1. While the study focused on broader effects of VARS on mRNA translation, it does not address if VARS alterations in the BRAFi-resistant melanomas could influence the mutant BRAF activated MAPK pathway itself, which has been shown in multiple studies, to be altered in BRAFi-resistant melanomas resulting in resistance. A simple western blotting showing the effect of VARS knockdown or upregulation on the protein levels of BRAF, MEK, Phospho-MEK, MAPK (ERK1/2) and Phospho-MAPK (ERK1/2) would help understand its potential specific effects on the MAPK pathway itself.

2. While I know that A375 and M395 cells are homozygous for BRAF-V600E, I do not know about the other cell lines in the study. This may be true for most readers. It is important to understand if there is heterozygosity in any of the cell lines in this study, and if BRAFi/MEKi treatment could potentially be selecting for BRAF-WT transcripts in the resistant cells, which could be happening at level of transcription or translation. This is especially important, as the cell lines are not consistently used in the experiments. Different cell lines were used for different experiments in the study.

3. On the same lines as point 2 above, I am intrigued to know if VARS upregulation may result in selective complete translation BRAF-WT transcripts at a higher rate than the mutant-BRAF transcripts. This could be important to know because after all, it is the Valine amino acid that is substituted out in BRAF-V600E mutants. Also, what is the possibility that VARS upregulation may result in mis-incorporation of Valine instead of Glutamic acid in the 600 aa position of BRAF?

4. The concentrations of BRAF and MEK inhibitors used in this study are too high. Vemurafenib at doses of 10 and 20 micro molar, and Trametinib at a dose of 1 micro molar could have non-specific effects. Also, the treatment times are not indicated in the nuclear fragmentation assays. The methods section suggests 24h, 48h and 72h. What times do the data represent? Additionally, different cell lines show sensitivity to BRAFi and MEKi at different times. For example, A375 may show significant nuclear fragmentation after 48h treatment, while others may show at 24h or 72h. So the combination of single time of treatment and extremely high doses of drugs makes the nuclear fragmentation data somewhat un-interpretable.

MTT based or Cell proliferation (growth) data showing dose-dependent effects on the cell lines used in the study could improve interpretation of the molecular data in the different cell lines. This could also help to address the concern about different cell lines for different experiments in point 2.

5. The findings of the association of VARS-mediated resistance with specific molecular pathways is interesting, and is concordant with findings from other studies. This may suggest a common denominator of resistance, which could be VARS-mediated, or a convergent effect of VARS-mediated and other mechanisms of resistance. Answering this question may be outside the scope of this study, but could be an interesting discussion point.

There are other smaller concerns:

6. The dosing schedule of Vemurafenib (daily once? twice? weekly once?) was not mentioned.
7. Fig 3d- while the text in line 157 says that VARS is upregulated in MM029 cell line, I don't see that compared to HSP90 control. Quantifying the VARS staining normalized to HSP90 would be helpful.
8. Line 162- Is the survival data for the melanoma patients from the TCGA? If yes, it should be indicated.
9. Its difficult to agree with authors interpretation of some of the nuclear fragmentation data- For example, in Fig 4d, VARS depletion did induce a small difference in melanoma survival in untreated compared to BRAFi/MEKi treated cells. The SD data does not indicate significance probably due to the very low percentage difference in fragmentation. Additionally, VARS-non depleted cells were also sensitized to BRAFi/MEKi, but at a lower level (Fig 4b and 4c).
10. Fig 7d,e,f- It will be good to show western blotting of MAPK and Phospho-MAPK in these figures.
11. Line 285- "VARS depletion led to a decrease of ribosomal content at the HADH transcripts". It's possible that this effect may be occurring during transcription, and not translation.

REFERENCES – are limited to a total of 70 for Articles, Resources, Technical Reports; and 40 for Letters. This includes references in the main text and Methods combined. References must be numbered sequentially as they appear in the main text, tables and figure legends and Methods and must follow the precise style of Nature Cell Biology references. References only cited in the Methods should be

numbered consecutively following the last reference cited in the main text. References only associated with Supplementary Information (e.g. in supplementary legends) do not count toward the total reference limit and do not need to be cited in numerical continuity with references in the main text. Only published papers can be cited, and each publication cited should be included in the numbered reference list, which should include the manuscript titles. Footnotes are not permitted.

Methods should be written concisely, but should contain all elements necessary to allow interpretation and replication of the results. As a guideline, Methods sections typically do not exceed 3,000 words. The Methods should be divided into subsections listing reagents and techniques. When citing previous methods, accurate references should be provided and any alterations should be noted. Information must be provided about: antibody dilutions, company names, catalogue numbers and clone numbers for monoclonal antibodies; sequences of RNAi and cDNA probes/primers or company names and catalogue numbers if reagents are commercial; cell line names, sources and information on cell line identity and authentication. Animal studies and experiments involving human subjects must be reported in detail, identifying the committees approving the protocols. For studies involving human subjects/samples, a statement must be included confirming that informed consent was obtained. Statistical analyses and information on the reproducibility of experimental results should be provided in a section titled “Statistics and Reproducibility”.

All Nature Cell Biology manuscripts submitted on or after March 21 2016 must include a Data availability statement at the end of the Methods section. For Springer Nature policies on data availability see <http://www.nature.com/authors/policies/availability.html>; for more information on this particular policy see <http://www.nature.com/authors/policies/data/data-availability-statements-data-citations.pdf>. The Data availability statement should include:

- Accession codes for primary datasets (generated during the study under consideration and designated as "primary accessions") and secondary datasets (published datasets reanalysed during the study under consideration, designated as "referenced accessions"). For primary accessions data should be made public to coincide with publication of the manuscript. A list of data types for which submission to community-endorsed public repositories is mandated (including sequence, structure, microarray, deep sequencing data) can be found here <http://www.nature.com/authors/policies/availability.html#data>.
- Unique identifiers (accession codes, DOIs or other unique persistent identifier) and hyperlinks for datasets deposited in an approved repository, but for which data deposition is not mandated (see here for details <http://www.nature.com/sdata/data-policies/repositories>).

- At a minimum, please include a statement confirming that all relevant data are available from the authors, and/or are included with the manuscript (e.g. as source data or supplementary information), listing which data are included (e.g. by figure panels and data types) and mentioning any restrictions on availability.
- If a dataset has a Digital Object Identifier (DOI) as its unique identifier, we strongly encourage including this in the Reference list and citing the dataset in the Methods.

We recommend that you upload the step-by-step protocols used in this manuscript to the Protocol Exchange. More details can found at www.nature.com/protocolexchange/about.

All imaging data should be accompanied by scale bars, which should be defined in the legend. Cropped images of gels/blots are acceptable, but need to be accompanied by size markers, and to retain visible background signal within the linear range (i.e. should not be saturated). The boundaries of panels with low background have to be demarked with black lines. Splicing of panels should only be considered if unavoidable, and must be clearly marked on the figure, and noted in the legend with a statement on whether the samples were obtained and processed simultaneously. Quantitative comparisons between samples on different gels/blots are discouraged; if this is unavoidable, it should only be performed for samples derived from the same experiment with gels/blots were processed in parallel, which needs to be stated in the legend.

Figures should be provided at approximately the size that they are to be printed at (single column is 86 mm, double column is 170 mm) and should not exceed an A4 page (8.5 x 11"). Reduction to the scale that will be used on the page is not necessary, but multi-panel figures should be sized so that the whole figure can be reduced by the same amount at the smallest size at which essential details in each panel are visible. In the interest of our colour-blind readers we ask that you avoid using red and green for contrast in figures. Replacing red with magenta and green with turquoise are two possible colour-safe alternatives. Lines with widths of less than 1 point should be avoided. Sans serif typefaces, such as

Helvetica (preferred) or Arial should be used. All text that forms part of a figure should be rewritable and removable.

The total number of Supplementary Figures (not including the “unprocessed scans” Supplementary Figure) should not exceed the number of main display items (figures and/or tables (see our Guide to Authors and March 2012 editorial <http://www.nature.com/ncb/authors/submit/index.html#suppinfo>; <http://www.nature.com/ncb/journal/v14/n3/index.html#ed>). No restrictions apply to Supplementary Tables or Videos, but we advise authors to be selective in including supplemental data.

GUIDELINES FOR EXPERIMENTAL AND STATISTICAL REPORTING

REPORTING REQUIREMENTS – To improve the quality of methods and statistics reporting in our papers we have recently revised the reporting checklist we introduced in 2013. We are now asking all life sciences authors to complete two items: an Editorial Policy Checklist (found here <https://www.nature.com/authors/policies/Policy.pdf>) that verifies compliance with all required editorial policies and a reporting summary (found here <https://www.nature.com/authors/policies/ReportingSummary.pdf>) that collects information on experimental design and reagents. These documents are available to referees to aid the evaluation of the manuscript. Please note that these forms are dynamic ‘smart pdfs’ and must therefore be downloaded and completed in Adobe Reader. We will then flatten them for ease of use by the reviewers. If you would like to reference the guidance text as you complete the template, please access these flattened versions at <http://www.nature.com/authors/policies/availability.html>.

We strongly recommend the presentation of source data for graphical and statistical analyses as a separate Supplementary Table, and request that source data for all independent repeats are provided when representative experiments of multiple independent repeats, or averages of two independent experiments are presented. This supplementary table should be in Excel format, with data for different figures provided as different sheets within a single Excel file. It should be labelled and numbered as one of the supplementary tables, titled “Statistics Source Data”, and mentioned in all relevant figure legends.

Author Rebuttal to Initial comments

- Rebuttal letter -
Manuscript #A51646
by El Hachem et al.

Nature Cell Biology editors discuss the referee reports in detail within the editorial team, including the chief editor, to identify key referee points that should be addressed with priority, and requests that are overruled as being beyond the scope of the current study. To guide the scope of the revisions, I have listed these points below. We are committed to providing a fair and constructive peer-review process, so please feel free to contact me if you would like to discuss any of the referee comments further.

We wish to thank the editors and the referees for their positive evaluation of our manuscript and their constructive comments. Please find enclosed a point-by-point response to the comments. The *key referee points* identified by the editorial team are referred below as “**Editor’s key points**”. All modifications introduced in the revised manuscript have been highlighted in **yellow** in the manuscript file.

Reviewers' Comments:

Reviewer #1:

Remarks to the Author:

In this manuscript, El-Hachem and colleagues find that melanoma cells that have acquired resistance to MAPK inhibition therapy upregulate expression of proteins enriched in valine amino acids. They further demonstrate that Valine tRNA and the VARS valine aminoacyl synthetase enzyme are upregulated in these cells. They go on to show that depletion of VARS sensitizes melanoma cells to MAPK inhibition therapy. They go on to identify the fatty acid oxidation gene HADH as a valine-enriched translationally upregulated protein that mediates resistance to MAPK inhibition. These findings reveal specific tRNA/charging enzyme modulation to underlie resistance to a widely clinically used therapeutic and extend in a meaningful way—with therapeutic significance and novelty—on previous studies that have demonstrated tRNA/charging enzyme modulation roles in cancer progression. The work is well done, novel, rigorous and addresses an important basic and clinically significant problem of therapeutic resistance. The manuscript is well-

written, concise and clear. I have only a few, yet critical recommendations below that need to be addressed prior to publication in NCB.

We thank the reviewer for her/his positive evaluation of our work. We addressed all the recommendations requested by the reviewer. Please see below our answers to each of the points raised.

1. The key finding of Valine tRNA modulation need to be validated by Northern, as this is central to the story and Northern analysis is the gold standard. (Editor's key points)

We assessed valine tRNA-CAC expression by Northern as validation of the previous approaches described in the manuscript in MM029 SENS & RES patient-derived lines (probes tRNA-Val-CAC: 5'-GAGGCGAACGTGATAACCACTACTACGGAAAC-3' and 5srRNA: 5'-CCTGCTTAGCTTCCGAGATCA-3' – Supplementary information for reviewers). This experiment is now added in main Fig. 2f, as well as in the main text. A representative experiment is shown, and a quantification of the replicates is illustrated in Supplementary Fig. 2d. These data validated that expression of valine tRNA-CAC is upregulated in RES melanoma lines, as compared to their SENS counterparts.

2. The in vivo finding in Fig. 4f of VARS depletion synergizing with Vemurafinib needs to be repeated with an independent shRNA to ensure lack of off-target effects, as this is a key and important discovery in the manuscript. The western blots showing HADH protein abundance reduction also needs to be repeated with an additional VARS shRNA. (Editor's key points)

We performed the requested experiments using the second, validated, shRNA targeting VARS (shVARS-2; already used in the first version of the manuscript).

First, we repeated the *in vivo* experiment in Fig. 4f. It has now been included in the manuscript in new Supplementary Fig. 4a. Strikingly, in line with our previous data, the results show that VARS depletion, using a second independent shRNA, re-sensitizes A375 resistant melanoma tumors to MAPK-based therapy.

Second, the western blots showing HADH protein abundance after depleting VARS with the second VARS shRNA are now provided in the manuscript in new Supplementary Fig. 7d. HADH protein expression was systematically decreased upon VARS depletion in the three tested melanoma lines, validating our conclusions.

3. The critical conclusion is that VARS upregulation and enhanced valine tRNA promote translation of HADH in a codon-dependent manner needs to be demonstrated using a codon-specific reporter. This is critical and will demonstrate codon-dependent effects and assuage future

nay-sayers who will claim that a direct mutagenesis experiment was not done to exclude non-direct effects on the HADH target gene. (Editor's key points)

We did not claim in the paper that down-regulation of proteins such as HADH after VARS depletion was caused by an effect on a specific valine codon. Rather, we show that VARS depletion affects the translation of transcripts enriched in all valine codons.

This is in line with observations made in previous work from colleagues - see for example *Thandapani et al, Nature 2022, DOI: 10.1038/s41586-021-04244-1*. There, they showed that Val-TAC is overexpressed in Leukemia, but they did not see clear enrichment of the cognate GTA or the wobble GTC codons over GTT and GTG codons in genes translationally down regulated upon VARS inhibition. In these conditions, doing a codon-specific reporter (i.e. using synonymous codons) will not add complementary information in the manuscript. In any case, we performed two different experiments to specifically address this comment.

First, we generated two different HADH mutants in which all the valine codons were replaced by either leucine (i.e. Flag-OE HADH V to L) or isoleucine (i.e. Flag-OE HADH V to Ile) codons (the other branched-chain amino acids and structurally closest to valine). These mutant constructs were then expressed in melanoma lines. Unfortunately, the newly generated mutants do not seem to be functional: 1/ they are much less expressed in cells; 2/ they migrate at a very different size as compared to the HADH-WT (i.e. Flag-OE HADH WT); and 3/ while the overexpression of HADH-WT greatly enhanced the FAO activity as expected, the expression of the two mutants failed to do so (Fig. reviewers 1a-b). Overall, this shows that systematically mutating all valine codons in HADH is very detrimental for its integrity and functionality.

Second, we also generated another reporter construct to assess valine codon-specific translation. We cloned 3x valine GTG codons upstream and in frame of the NanoLucPEST cDNA. This is a strategy we previously used when we studied the dependency of AAA-GAA or CAA codons towards wobble uridine tRNA modification enzymes (*Rapino et al, 2021; DOI: 10.1038/s41467-021-22254-5, cfr Fig. 1a*). We expressed this construct in melanoma cells and we depleted or not VARS (shCTRL, shVARS-1 and shVARS-2). Our data show that VARS depletion leads to reduced NanoLuc signal when NanoLuc is preceded with 3xGTG codons (Fig. Reviewers 1c), the most used valine codon. Nevertheless, we observed a strong and systematic difference in NanoLuc signal in the control cells between the WT-NanoLuc and the 3xGTG-Nanoluc constructs (compare lines 1 and 4). We do not explain this difference, but we believe that it prevents us to make any clear and unambiguous conclusions regarding the codon-specific regulation mentioned by the referee. For the sake of clarity, we **made sure to avoid any confusing statement regarding the codon-specificity** (i.e. specific codon usage) in the manuscript.

As a final remark, although we cannot formally exclude that other mechanisms may be conjointly involved, we would like to emphasize that we provide in the manuscript a series of evidence (i.e. polysome sequencing, ribosome sequencing and ribosome IP experiment) showing that VARS

regulates HADH and other target genes mainly at the level of mRNA translation, and not through another indirect mechanism.

Figure Reviewer #1

Figure legend

a, HADH WT (Flag-OE-HADH WT) construct or HADH valine to leucine mutant (Flag- OE-HADH V to L) or HADH valine to isoleucine mutant (Flag- OE-HADH V to Ile) were overexpressed in M395 cells.

western blot is shown and HSP90 is used for normalization. **b**, as in a but fatty acid oxidation activity was measured and quantified. **c**, NanoLuc preceded or not by 3xGTG codons in frame was cloned in a lentivirus expressing vector and expressed in M395 cells which were then depleted of VARS. Luminescence signal was measured and plotted. Two-way ANOVA was performed. Tukey's multiple comparisons test are indicated in the figure.

Reviewer #2:

Remarks to the Author:

This work describes the role of a specific aminoacyl-tRNA synthetase in the therapy resistance of melanoma. Aminoacyl-tRNA synthetases have been studied for a long time as house-keeping genes that are required for protein synthesis and cell growth. Recently, numerous studies have shown that specific tRNA synthetases play various roles in different types of cancer, either as promoters or suppressors. The cancer-specific roles of AARS are generally derived from their effects on cognate tRNA charging that are important to fine-tune translation of specific genes required for relevant cancer processes such as metastasis and others. This work shows that valine tRNA synthetase (VARS) plays a promotional role in the codon-dependent translational reprogramming induced by resistance to targeted MAPK therapy in melanoma. When melanoma becomes drug resistant, their proteomes are biased toward the usage of valine, which is also accompanied by valine-tRNA and VARS overexpression. The authors also identify specific pathways involved in the resistance, such as an enzyme in fatty acid oxidation. They also show that resistant melanoma cultures are addicted to fatty acid oxidation and this enzyme for survival upon drug treatment. This work adds the growing literature on specific tRNA acting enzymes in specific cancers, but with the exiting new twist on finding the specific metabolic pathways required, as well as finding an essential house-keeping gene as cancer target in this specific context.

We thank the reviewer for her/his positive assessment and her/his enthusiasm. We provide here below a systematic point-by-point responses to her/his comments.

1. As a study dealing with tRNA synthetase and tRNA expression and charging, the biochemical characterization does not reflect the state-of-the-art of the field. The authors used something like hydro-seq from 2017 which relies on sequencing fragmented tRNAs. The authors seem to have performed demethylase treatment to remove m1A and m3C, but the efficiency of this step was not described. In any case, multiple new tRNA-seq procedures have been published since 2020 which can provide more precise information on tRNA expression. (Editor's key points)

There is a misunderstanding of the procedure used in this manuscript. While tRNA expression profiling is still under continuous improvement, the tRNA-seq used here is a significant

enhancement over hydro-tRNA-seq by including demethylation and major improvement over regular small RNA sequencing for tRNAs:

*Regular small RNA sequencing for tRNA: there is no demethylation step and no partial hydrolysis to overcome tRNA tight folding. Reads are therefore poorly mappable.

*Hydro-tRNA-seq: this includes disruption of tRNA fold by hydrolysis, but no demethylation.

In our experiment, both demethylation and tRNA fold disruption were included.

For tRNA sequencing, tRNAs are isolated from the total RNA by gel electrophoresis in the size range of 60-100 nt (tRNA fragments are excluded). The isolated tRNAs are then demethylated and partially hydrolyzed. The pre-treated tRNAs are converted to cDNA library using the Small RNA Library Prep Set Kit from Illumina.

Therefore, the sequencing methodology used on this manuscript respects all the standards and can be trusted: variations in specific tRNA expression (up - no change - down) were extensively confirmed by subsequent tRNA-centric qRT-PCR in different patient-derived melanoma cultures. We also confirmed the upregulation of valine tRNA-CAC by northern blot (see below; new Fig. 2f; Supplementary Fig. 2d)

Fig. 2: The inadequate sequencing likely reflects the fact that qRT-PCR shows >4-fold change, but sequencing shows maybe 1.5-fold change in tRNA^{Val} expression. The authors should validate tRNA^{Val} expression results by Northern blot, not by qRT-PCR because of the differences in RT-interfering modifications. (Editor's key points)

High-throughput and more specific methods may indeed give different numbers in the extend of RNA expression fold. This is also largely seen when studying mRNA expression using different methods (RNAseq vs qRT-PCR).

As requested, and previously discussed, we assessed valine tRNA-CAC expression by Northern as validation of the previous approaches described in the manuscript in MM029 SENS & RES patient-derived lines (probes tRNA-Val-CAC: 5'-GAGGCGAACGTGATAACCACTACTACGGAAAC-3' and 5srRNA: 5'-CCTGCTTAGCTTCCGAGATCA-3'). This experiment is now added in main Fig. 2f. A representative experiment is shown, and a quantification of the replicates is illustrated in Supplementary Fig. 2d. These data validated that expression of valine tRNA-CAC is upregulated in RES melanoma lines, as compared to their SENS counterparts.

Of note, the purpose of qRT-PCR experiments shown in the manuscript is to assess if the changes identified using one line are also seen across different lines. Strikingly, the data convincingly show that the observations are reproductively seen across different patient-derived melanoma lines.

2. Fig. 3, 4: tRNA charging was measured using a complicated RT-PCR method without adequate controls. The standard method to directly and unambiguously measure tRNA charging is by acid-denaturing gel electrophoresis. There are now non-radioactive methods for tRNA Northern blots using biotinylated probes for charging measurements, no radioactivity needed. (Editor's key points)

We heard the comment of the reviewer. To address this concern, we have extensively discussed this question with expert scientists in the proposed methodologies.

#1. The RT-PCR coupled to periodate oxidation method used in our manuscript to assess tRNA charging has been largely used and extensively validated in the literature in seminal papers such as in (non-exhaustive list):

Thandapani et al, Nature 2022, DOI : 10.1038/s41586-021-04244-1

Pavlova et al, eLife 2020, DOI : 10.7554/eLife.62307

Loayza-Puch et al, Nature 2016, DOI: 10.1038/nature16982

Dittmar et al, EMBO Rep. 2005, DOI : 10.1038/sj.embor.7400341

We have added **additional controls** for the RT-PCR experiments shown in the manuscript. First, we now show that the charging state of isoleucine tRNAs remain unchanged in SENS and RES cells (main Fig. 3c), as well as in VARS depleted cells (main Fig. 4a). Second, as an additional demonstration that the methodology adequately reports measurement of the valine aminoacylation activity, we have also measured the aminoacylation activity in SENS melanoma lines after VARS overexpression. This experiment shows that VARS overexpression leads to enhanced aminoacylation of valine tRNAs, while it does not affect aminoacylation of isoleucine tRNAs (Fig. reviewers 2a). These evidences provide strong arguments demonstrating that this the proposed methodology is adequate to measure VARS activity in the different conditions. Hereby, we would like to thank Dr. Palaniraja Thandapani (MD Anderson Cancer Center) for his support in performing these experiments. He is now in the list of authors.

#2. As requested by the reviewer, we also sought to measure the valine tRNA charging by acid-denaturing gel electrophoresis. Because we are not expert in this matter (protocol and interpretation), we performed these experiments in collaboration with Dr. Ivan Tarassov, a recognized expert from the Strasbourg GMGM Research institute. We wish to thank him for his support and the extensive discussions. Acid-denaturing gel electrophoresis experiments were carried out using SENS or RES melanoma lines. The data show that the nearly all the valine tRNA-CAC is acylated in SENS as well as in RES cells (Fig. Reviewers 2b). Indeed, only one band is detected in both SENS or RES conditions and migrates slower than the deacylated controls (which are provided in lines 1 and 3) confirming that the band detected are fully aminoacylated tRNAs. Because no valine tRNA-CAC was ever detected in its deacylated form (in

lines 2 and 4, even after very long exposure), we could not calculate acylated/non-acylated ratio in these conditions. Importantly, while the same amount of tRNA was loaded in the SENS and the RES lines, the bands detected in RES cells were always much more intense, indicating that the absolute amount of valine tRNA-CAC charged is higher in RES melanoma lines. This is consistent with our data presented in figure 2 showing by an independent northern blot detection that the absolute amount of val-tRNA is higher in RES. This suggests that RES melanoma lines, which harbor an elevated VARS expression, an enhanced quantity of valine tRNA-CAC expression, aminoacylate more tRNA-CAC, as indicated with other methods.

Figure Reviewer #2

Figure legend

a, Val-MAC tRNA and Ile-RAT tRNA aminoacylation analysis of M395 SENS after VARS overexpression (empty vector was used as control; mean \pm SEM of $n=2$ independent replicates). **b**, Acid gel to measure aminoacylation levels of Val-tRNA^{Val} in SENS or RES melanoma lines.

3. Fig. 5: how do cells still do well in translation after severe knockdown of an essential house-keeping gene like VARS? I am surprised to see that the VARS depletion does not trigger ISR. I am also surprised to see that ribosome does not seem to stall at Val codons in the A site. Does this mean the polysome is still fully loaded with charged tRNA^{Val} despite the decrease of charged tRNA^{Val} levels in the total tRNA? (Editor's key points)

We thank the reviewer for this comment. As her/him, we were also puzzled to observe that VARS knock-down only slightly impacts translation. Several complementary approaches were performed to assess mRNA translation after VARS KD. All of them converged to the same conclusion: VARS KD does not globally affect translation; only subsets of mRNA, enriched in

valine codons, are affected in their translation. Moreover, despite several attempts using different RiboSeq protocols, we could never observe any specific stalling of ribosomes at valine codons in the A site. It indicates that cells can cope with efficient VARS KD for their translation. Different mechanisms could be implicated in this phenomenon; we have extensively discussed them in the discussion section of the paper (lines 291-312):

“(…). Our data showed that VARS knockdown does not significantly compromise melanoma cellular survival, nor tumor growth in mice, unless cells or tumors are treated with MAPK therapy. As such, VARS activity protects melanoma from MAPK-therapy, at least through promoting HADH translation and FA oxidation cellular activity. Importantly, VARS depletion does not completely abolish the cellular valine aminoacylation activity, suggesting that remaining VARS activity may maintain melanoma survival in untreated conditions. Consistently, we do not observe any global changes in mRNA regulation upon VARS depletion. Other aminoacyl-tRNA synthetases may compensate the limited VARS activity. Although aminoacyl-tRNA synthetase are highly conserved enzymes, mistranslation can occur with amino acids sharing similar structure, leading to generation of nascent proteins with mis-incorporated amino acids. For example, isoleucine aminoacyl-tRNA synthetase binds isoleucine, but can also accommodate valine to tRNA^{Ile}, although at a rate of 1/200. Similarly, VARS can also load threonine to tRNA^{Val} with lower affinity. Studies on cross-aminoacylation showed that mitochondrial aminoacyl transferases may be active at corresponding cytoplasmic tRNAs. However, these hypotheses remain to be tested. Of note, VARS knockdown in melanoma cells is not associated with any change in expression of other aminoacyl-tRNA synthetases. Another intriguing possibility is that specific codon reassignment may take place in VARS knockdown cells. This phenomenon was previously described as the codon capture theory. Recently, tryptophan aminoacyl-tRNA synthetase was shown to drive tryptophan-to-phenylalanine codon reassignment in cells deprived of tryptophan. These hypotheses, which may explain how cells compensate the lack of VARS activity, will be the subject of future studies. (…)”

Reviewer #3:

Remarks to the Author:

This study by Najla El-Hachem et al is a well-written, interesting and extensive investigation of the role of Valine amino acyl tRNA synthetase (VARS) in BRAF inhibitor (BRAFi)-resistant melanomas. My comments are specifically focused on interpretation of the cellular and molecular effects of VARS alterations in the melanoma cells, tumors and clinical samples.

The discovery of the BRAF-V600E mutation as a driving force in melanoma oncogenesis and growth led to the development of BRAF-V600 mutant specific targeted inhibitors and MEK inhibitors targeting the BRAF-activated MAPK pathway. Subsequent clinical studies showed dramatic efficacy of these drugs at low nanomolar doses, but also showed that the efficacy is short-lived due to development of resistance. Many studies have since shown different

mechanisms of resistance to these inhibitors. Some results from this study parallel other studies which have shown the role of metabolic pathways in resistance, for example the role of Fatty acid oxidation and its key enzymes. However, the mechanisms of these broad pathway-dependent resistances are poorly defined. It is possible that VARS may play an important role in the generation of such resistance. While this study is compelling in that context, there are several questions and experimental concerns as described below. Addressing these concerns could potentially improve the study.

We thank the reviewer for her/his evaluation. We have addressed all the concerns raised by the reviewer and updated the manuscript accordingly. Please see our systematic responses below.

1. While the study focused on broader effects of VARS on mRNA translation, it does not address if VARS alterations in the BRAFi-resistant melanomas could influence the mutant BRAF activated MAPK pathway itself, which has been shown in multiple studies, to be altered in BRAFi-resistant melanomas resulting in resistance. A simple western blotting showing the effect of VARS knockdown or upregulation on the protein levels of BRAF, MEK, Phospho-MEK, MAPK (ERK1/2) and Phospho-MAPK (ERK1/2) would help understand its potential specific effects on the MAPK pathway itself. (Editor's key points)

We performed the requested western blot analysis in three $BRAF^{V600E}$ mutated melanoma SENS or RES lines (i.e. A375-M395-MM029). They have been added in the manuscript in Supplementary Fig. 7h-i. The data show that modulation of VARS expression does not impact BRAF expression, nor MAPK pathway activation (i.e. pMEK and pERK) in the tested $BRAF^{V600E}$ mutated melanoma SENS or RES lines. A sentence has been added in the discussion (lines 325-327).

2. While I know that A375 and M395 cells are homozygous for BRAF-V600E, I do not know about the other cell lines in the study. This may be true for most readers. It is important to understand if there is heterozygosity in any of the cell lines in this study, and if BRAFi/MEKi treatment could potentially be selecting for BRAF-WT transcripts in the resistant cells, which could be happening at level of transcription or translation. This is especially important, as the cell lines are not consistently used in the experiments. Different cell lines were used for different experiments in the study. (Editor's key points)

We thank the reviewer for this important comment. We performed the requested analyses to answer this concern.

First, we performed DNA sequencing of the lines after amplification of the *BRAF* exon 15 (which encloses the mutation; V=GTG into E=GAG) using the following primers: 5'-TCATAATGCTTGCTCTGATAGGA-3' (forward) and 5'-GGCCAAAATTTAATCAGTGGA-3' (reverse) and using high fidelity polymerase (as previously described in Zhao et al., 2019¹). All

sequences were analyzed after sequencing with both the forward and the reverse primers. Results are illustrated in Fig. Reviewers 3a. They show that lines A375, MM395 and MM029, which are mostly used throughout the paper, are homozygous for the $BRAF^{V600E}$ mutation while the lines MM074, MM383 and MM099 are heterozygous for the $BRAF^{V600E}$ mutation.

Next, we assessed the expression of $BRAF^{WT}$ and $BRAF^{V600E}$ transcripts in the 3 heterozygous RES lines after treatment using amplicon sequencing after RNA extraction and RT-PCR and using the following primers:

1/ Forward overhang:

5'TCGTCGGCAGCGTCAGATGTGTATAAGAGACAG-[locus- specific sequence]

5'TCGTCGGCAGCGTCAGATGTGTATAAGAGACAGTCATAATGCTTGCTCTGATAGGA-3'

2/ Reverse overhang:

5'GTCTCGTGGGCTCGGAGATGTGTATAAGAGACAG-[locus- specific sequence]

5'GTCTCGTGGGCTCGGAGATGTGTATAAGAGACAGGGCCAAAATTTAATCAGTGGA-3'

As for quantification of the sequencing data, we generated a standard curve using DNA from cells homozygous for $BRAF^{V600E}$ ($BRAF^{V600E}/BRAF^{WT}$: 100/0), heterozygous ($BRAF^{V600E}/BRAF^{WT}$: 50/50), or a mix of the two ($BRAF^{V600E}/BRAF^{WT}$: 75/25).

We next assessed the impact of treatment on $BRAF^{WT}$ and $BRAF^{V600E}$ transcripts expression in MM099 and MM383 SENS cells and their resistant counterparts were treated in culture (with BRAFi/MEKi). No significant difference was observed in the abundance of the $BRAF^{WT}$ transcripts (nor in $BRAF^{V600E}$ transcripts) as illustrated in Fig. Reviewers 3b.

Finally, we also assessed whether the $BRAF^{WT}$ and $BRAF^{V600E}$ mRNA translation may be impacted by treatment by MAPKi in resistant melanoma. MM074 cells were treated with 5 and 10 μ M of BRAFi (i.e. vemurafenib) and a polysome profiling experiment was performed. mRNAs from input and polysome fraction were extracted and the abundance of the $BRAF^{WT}$ and $BRAF^{V600E}$ transcripts was evaluated as previously using amplicon sequencing (Fig. Reviewers 3c). Strikingly no significant difference in the expression or the ratio of the two transcripts was observed in the total abundance of the transcripts or the polysome fractions (i.e. translation of the transcripts).

Overall, these data, combined with data of the previous answers (Supplementary Fig. 7h-i), show that nor the expression of BRAF (WT or mutated; mRNA expression – mRNA translation and protein expression), nor the activity of the downstream MAPK pathways are affected by modulation of VARS in melanoma.

Figure Reviewer #3

Figure legend

a, Sanger sequencing analysis of *BRAF* exon 15 in M395, A375, MM029, MM074, MM099 and MM383 melanoma cells. **b**, Quantification of *BRAF* WT and *BRAF* V600E transcripts by amplicon sequencing on MM099 and MM383 SENS and RES treated cells. **c**, Quantification of *BRAF*^{WT} and *BRAF*^{V600E} translation and transcription by amplicon sequencing after polysome profiling on MM074 RES treated or not with the indicated amount of Vemurafenib.

3. On the same lines as point 2 above, I am intrigued to know if VARS upregulation may result in selective complete translation BRAF-WT transcripts at a higher rate than the mutant-BRAF transcripts. This could be important to know because after all, it is the Valine amino acid that is substituted out in BRAF-V600E mutants. Also, what is the possibility that VARS upregulation may result in mis-incorporation of Valine instead of Glutamic acid in the 600 aa position of BRAF? (Editor's key points)

VARS overexpression experiments were performed using lines homozygous for *BRAF*^{V600E}.

Regarding the mis-incorporation, we cannot fully rule out this intriguing possibility. Nevertheless, we looked at our proteomics data, with the help of Pr. Reuven Agami (NKI, NL): we did not see any valine substituents, nor frameshifting; also overall the levels of valine in all sample proteomics are similar.

4. The concentrations of BRAF and MEK inhibitors used in this study are too high. Vemurafenib at doses of 10 and 20 micro molar, and Trametinib at a dose of 1 micro molar could have non-specific effects.

Also, the treatment times are not indicated in the nuclear fragmentation assays. The methods section suggests 24h, 48h and 72h. What times do the data represent?

Additionally, different cell lines show sensitivity to BRAFi and MEKi at different times. For example, A375 may show significant nuclear fragmentation after 48h treatment, while others may show at 24h or 72h. So the combination of single time of treatment and extremely high doses of drugs makes the nuclear fragmentation data somewhat un-interpretable. (Editor's key points)

MTT based or Cell proliferation (growth) data showing dose-dependent effects on the cell lines used in the study could improve interpretation of the molecular data in the different cell lines. This could also help to address the concern about different cell lines for different experiments in point 2.

We heard this important comment. The main figure panels 4b, c, d and e were systematically updated with lower doses of inhibitors as requested by the reviewer.

Below some references showing the use of similar doses:

Rapino et al, 2018, Nature, DOI : 10.1038/s41586-018-0243-7

Cerezo et al 2016, Cancer Cell, DOI : 10.1016/j.ccell.2016.04.013

Dratkiewicz et al 2019, DOI: 10.3390/ijms21010113

Wang et al 2018, Cell, DOI: 10.1016/j.cell.2018.04.012

Also, we clarified the times used in the cell death data in the methods section: 24h for M395, MM029, MM383 and 72h for A375 cells. We apologize for the confusion.

5. The findings of the association of VARS-mediated resistance with specific molecular pathways is interesting and is concordant with findings from other studies. This may suggest a common denominator of resistance, which could be VARS-mediated, or a convergent effect of VARS-mediated and other mechanisms of resistance. Answering this question may be outside the scope of this study, but could be an interesting discussion point.

We thank the reviewer for this comment. This is indeed an intriguing possibility that falls beyond the scope of our study. Nevertheless, as suggested by the reviewer, we have added a sentence in the discussion section of the manuscript (lines 332-334).

There are other smaller concerns:

6. The dosing schedule of Vemurafenib (daily once? twice? weekly once?) was not mentioned.

We apologize for this confusion. Mice were treated every two days once with vemurafenib. This has been added in the corresponding methods section (line 556 of the revised manuscript).

7. Fig 3d- while the text in line 157 says that VARS is upregulated in MM029 cell line, I don't see that compared to HSP90 control. Quantifying the VARS staining normalized to HSP90 would be helpful

The requested quantifications have been added in Supplementary Fig. 3b, as requested. We observed a systematic upregulation of VARS in RES cultures.

8. Line 162- Is the survival data for the melanoma patients from the TCGA? If yes, it should be indicated.

We confirm that the survival data were acquired from GEPIA, which uses TCGA, this is already mentioned in the text, we made sure that no confusion remains in the manuscript (line 149).

9. It's difficult to agree with authors interpretation of some of the nuclear fragmentation data- For example, in Fig 4d, VARS depletion did induce a small difference in melanoma survival in untreated compared to BRAFi/MEKi treated cells. The SD data does not indicate significance probably due to the very low percentage difference in fragmentation. Additionally, VARS-non depleted cells were also sensitized to BRAFi/MEKi, but at a lower level (Fig 4b and 4c).

Our statement is the following (at line 163 in the revised manuscript): “*Surprisingly, VARS depletion did not **significantly** affect melanoma cells survival in untreated conditions*”. Although we agree with the reviewer that VARS depletion may induce a small difference in melanoma survival in untreated conditions (always less than 10% cell death in all tested melanoma cells), depending on the cells studied, the data did not reach significance. To avoid any confusing statement, we amended the discussion section as follows (line 292): “*Our data showed that VARS knockdown does not **significantly** compromise melanoma cellular survival (...)*”.

10. Fig 7d,e,f- It will be good to show western blotting of MAPK and Phospho-MAPK in these figures.

This point was also raised by another reviewer. We indeed assessed BRAF expression and MAPK pathway activation upon modulation of VARS (OE or KD). This is now shown in Supplementary Fig. 7h-i.

11. Line 285- “*VARS depletion led to a decrease of ribosomal content at the HADH transcripts*”. *It's possible that this effect may be occurring during transcription, and not translation.*

We cannot fully exclude that a small part of the effect may be due to differences in *HADH* mRNA stability or expression. Nevertheless, using various approaches, our data demonstrate that VARS regulates *HADH* at the level of translation, as validated by ribosome IP experiments (see Supplementary Fig. 7e). Therefore, our set of data support our statement that remains correct.

To all reviewers

We wish to mention that we have designed a model figure (now shown in Supplementary Figure 8) to illustrate our findings.

Reference:

1. Zhao, J. *et al.* Comparison of diagnostic methods for the detection of a BRAF mutation in

papillary thyroid cancer. *Oncol. Lett.* **17**, 4661–4666 (2019).

Decision Letter, first revision:

Our ref: NCB-A51646A

11th March 2024

Dear Dr. Close,

Thank you for submitting your revised manuscript "Valine aminoacyl-tRNA synthetase promotes therapy resistance in melanoma" (NCB-A51646A). It has now been seen by the original referees and their comments are below. The reviewers find that the paper has improved in revision, and therefore we'll be happy in principle to publish it in Nature Cell Biology, pending minor revisions to satisfy the referees' final requests and to comply with our editorial and formatting guidelines.

Thank you again for your interest in Nature Cell Biology Please do not hesitate to contact me if you have any questions.

Sincerely,

Zhe Wang, PhD
Senior Editor
Nature Cell Biology

Tel: +44 (0) 207 843 4924
email: zhe.wang@nature.com

Reviewer #1 (Remarks to the Author):

The authors have conducted the northern and secondary shRNA experiments I requested to my satisfaction and the manuscript is much improved. They could not conduct codon mutagenesis experiments to assess the direct regulation of the target gene by the tRNA due to protein stability effects when mutating the Valine codons to Isoleucine or Leucine. I understand the difficulty in these

experiments and would support the publication of the manuscript so long as the authors describe in the results and discussion section the caveat that the effects could in part be non-tRNA/charging enzyme dependent.

Reviewer #2 (Remarks to the Author):

The authors adequately addressed my comments in good faith.

Reviewer #3 (Remarks to the Author):

The authors of this study have addressed all my concerns using experimental evidence, language corrections and citations of appropriate literature.

Author Rebuttal, first revision:

- Rebuttal letter -
Manuscript #A51646
by *El Hachem et al.*

Reviewer #1 (Remarks to the Author):

The authors have conducted the northern and secondary shRNA experiments I requested to my satisfaction and the manuscript is much improved. They could not conduct codon mutagenesis experiments to assess the direct regulation of the target gene by the tRNA due to protein stability effects when mutating the Valine codons to Isoleucine or Leucine. I understand the difficulty in these experiments and would support the publication of the manuscript so long as the authors describe in the results and discussion section the caveat that the effects could in part be non-tRNA/charging enzyme dependent.

We thank the reviewer for his positive evaluation of our revision work.

We have included the following sentence at line 314 of the revised manuscript.

“The possibility that VARS may promote resistance through a non-tRNA dependent mechanism cannot be fully excluded”

Reviewer #2 (Remarks to the Author):

The authors adequately addressed my comments in good faith.

We thank the reviewer for his positive evaluation of our revision work.

Reviewer #3 (Remarks to the Author):

We thank the reviewer for his positive evaluation of our revision work.

Final Decision Letter:

Dear Dr Close,

I am pleased to inform you that your manuscript, "Valine aminoacyl-tRNA synthetase promotes therapy resistance in melanoma", has now been accepted for publication in Nature Cell Biology.

Please note that *Nature Cell Biology* is a Transformative Journal (TJ). Authors may publish their research with us through the traditional subscription access route or make their paper immediately open access through payment of an article-processing charge (APC). Authors will not be required to make a final decision about access to their article until it has been accepted. Find out more about Transformative Journals

If you have not already done so, we strongly recommend that you upload the step-by-step protocols used in this manuscript to protocols.io (<https://protocols.io>), an open online resource that allows researchers to share their detailed experimental know-how. All uploaded protocols are made freely

available and are assigned DOIs for ease of citation. Protocols and Nature Portfolio journal papers in which they are used can be linked to one another, and this link is clearly and prominently visible in the online versions of both. Authors who performed the specific experiments can act as primary authors for the Protocol as they will be best placed to share the methodology details, but the Corresponding Author of the present research paper should be included as one of the authors. By uploading your Protocols onto protocols.io, you are enabling researchers to more readily reproduce or adapt the methodology you use, as well as increasing the visibility of your protocols and papers. You can also establish a dedicated workspace to collect your lab Protocols. Further information can be found at <https://www.protocols.io/help/publish-articles>.

With kind regards,

Zhe Wang, PhD
Senior Editor
Nature Cell Biology

Tel: +44 (0) 207 843 4924
email: zhe.wang@nature.com
